# Antarctic contribution to future sea level from ice shelf basal melt as constrained by ice discharge observations

Eveline C. van der Linden[1], Dewi Le Bars[1], Erwin Lambert[1], and Sybren Drijfhout[1,2]

[1]Royal Netherlands Meteorological Institute, Utrechtseweg 297, 3731 GA, De Bilt, The Netherlands
[2]Institute for Marine and Atmospheric Research Utrecht, Department of Physics, Utrecht University, Princetonplein 5, 3584 CC, Utrecht, The Netherlands

**Correspondence:** Eveline C. van der Linden (linden@knmi.nl)

**Abstract.** Antarctic mass loss is the largest contributor to uncertainties in sea level projections on centennial timescales. In this study we aim to constrain future projections of the contribution of Antarctic dynamics by using ice discharge observations. The contribution of Antarctica's ice discharge is computed with ocean thermal forcing from 14 earth system models (ESMs) and linear response functions (RFs) from 16 ice sheet models for three shared socio-economic pathway (SSP) scenarios. New compared to previous studies, basal melt sensitivities to ocean temperature changes were calibrated on four decades of observed ice discharge changes rather than using observation-based basal melt sensitivities. Calibration improved historical performance, but did not reduce the uncertainty in the projections. The results show that even with calibration the acceleration during the observational period is underestimated for the Amundsen region, indicating missing physics. Also the relative contribution of the Amundsen region is underestimated. The Amundsen contribution and sea level acceleration are improved by choosing an Amundsen-specific calibration (rather than Antarctic-wide), quadratic basal melt parameterisation (rather than linear) and thermal forcing near the ice shelf base (rather than the deepest layer above the continental shelf). With these methodological choices we arrive at a median dynamic sea level contribution of 0.12 m for SSP1-2.6, 0.14 m for SSP2-4.5 and 0.17 m for SSP5-8.5 in 2100 relative to 1995-2014, sitting in between projections of previous multi-model studies (ISMIP6 emulator and LARMIP-2). Our results show that constraining the basal melt parameterisation on Amundsen ice discharge rather than applying the median basal melt sensitivities used in LARMIP-2 and the mean Antarctic distribution of ISMIP6 leads to higher sea level contributions. However, differences in basal melt sensitivities alone cannot explain the differences in our projections compared to emulated ISMIP6 and LARMIP-2. We conclude that uncertainties associated with ESMs and ice sheet models affect the projected sea level contribution more than our methodological choices in the calibration and basal melt computation.

## 1 Introduction

Sea level rise poses an increasing threat to densely populated coasts and deltas worldwide (Hinkel et al., 2014). Even if the 1.5 degree target of the Paris Agreement is met, global mean sea level will rise several meters in the longer term (Clark et al., 2016; Fox-Kemper et al., 2021). At present, a global acceleration of sea level rise is visible in satellite measurements and the sea level is already rising more than twice as fast as the average rate over the twentieth century (Nerem et al., 2018; Dangendorf et al., 2019).

Mass loss from land ice (ice sheets and glaciers) is currently accelerating and is now (over the period 2006–2018) the largest contributor to the global mean sea level rise (Fox-Kemper et al., 2021). Antarctic ice sheet (AIS) mass loss has tripled over the last decade (Shepherd et al., 2018), which can be mainly attributed to increased ice discharge in the Amundsen Sea (Rignot et al., 2019). Models and geological data indicate that the AIS will cause most of the sea level rise over thousands of years (Bamber et al., 2019). The degree of acceleration of future sea level changes is mainly determined by the dynamic contribution of the AIS. The underlying processes are 1) increased melting of ice shelves by warmer ocean water (basal melt) and 2) increased calving (iceberg formation) triggered by basal melt and/or surface melt (Rignot and Jacobs, 2002; van den Broeke, 2005; Pritchard et al., 2012; Liu et al., 2015).

Melt of Antarctic land ice is the largest contributor to uncertainties on centennial timescales (van de Wal et al., 2019; Palmer et al., 2020). It is important to gain a better understanding of the many uncertainties about the Antarctic contribution to sea level rise that exist and to reduce these uncertainties when possible to support adaptation planning (Haasnoot et al., 2020). Uncertainties associated with the Antarctic contribution to sea level rise appear to be increasing since more and more models and processes are included in the uncertainty assessments. Using similar methodologies to each other, the estimated Antarctic contribution in Levermann et al. (2020) shows increased uncertainty compared to its previous study (Levermann et al., 2014) and expert judgment assessments of Bamber et al. (2019) give higher uncertainties than before (Bamber and Aspinall, 2013). To address this issue, our study aims to gain more insight in the Antarctic contribution to, and uncertainties in, future sea level changes and provides directions for reducing these uncertainties.

Future projections of Antarctic mass loss are based on modelling studies, in which ice sheet models are used as a standalone unit and forcing is provided by earth system models (ESMs). Over the last decade, ice sheet modelling has advanced from single model studies to model intercomparison projects (MIPs). In these projects, earth system modelling and ice sheet modelling are combined to make projections of land ice. The Ice Sheet MIP for CMIP6 (ISMIP6) (Nowicki et al., 2016) and Linear Antarctic Response MIP (LARMIP-2) (Levermann et al., 2020) are currently used as one basis for projections of the Antarctic land ice evolution (Fox-Kemper et al., 2021). ISMIP6 (Seroussi et al., 2020) provides process-based projections of the sea level contribution of the AIS based on a variety of ice sheet models that are forced by atmosphere and ocean output from ESMs. ISMIP6 made a selection of six ESMs based on two main criteria. The first criterion is based on their performance in reproducing the mean state of the current climate (atmosphere and ocean) near Antarctica, but did not include trends. The second criterion ensures that the ESM selection includes a diversity of warming rates over the 21st century so that the uncertainty-range in projections is captured (Barthel et al., 2020; Nowicki et al., 2020). One risk of this selection process is that models with a relatively bad performance over the historical period in terms of trends could have been chosen. In ISMIP6 basal melt is calibrated on basal melt observations with two options for calibration: the mean AIS and Pine Island's grounding line (Jourdain et al., 2020). LARMIP-2 focuses on ice sheet mass loss due to ice shelf basal melt (Levermann et al., 2014, 2020). In that study, the temperature melt-relation is parameterised with a linear dependency on thermal forcing. ISMIP6 and LARMIP-2 have thirteen ice sheet models in common and are primarily based on the CMIP5 ESMs and scenarios (RCPs) as forcing. Payne et al. (2021) demonstrate that the estimated AIS mass loss in ISMIP6 models with CMIP6 forcing is similar compared to using CMIP5 forcing. Edwards et al. (2021) estimated probability distributions for projections under the SSP

scenarios based on CMIP6 ESMs, by using statistical emulation of the ISMIP6 ice sheet models. The main projections of the sea level contribution from the AIS in the 6th Assessment Report of the Intergovernmental Panel on Climate Change (IPCC AR6; Fox-Kemper et al., 2021, their Tab. 9.3) are based on a combination of the projections of LARMIP-2 and emulated ISMIP6.

Our study follows LARMIP-2 to account for the sensitivity of ice sheet models to climate change by using linear response functions (RF) of ice sheet models. The LARMIP-2 RFs were obtained by prescribing for five Antarctic regions an immediate change in basal melt of the ice shelves and simulating the resulting ice loss with the ice sheet model. The changes in the volume above flotation of the ice sheet are then calculated to obtain the sea-level equivalent ice loss. In this way a relationship between basal melt and the related contribution to sea level is obtained for each region: the linear response function. Additionally, a relationship between thermal forcing and basal melt is used to compute basal melt from ocean temperatures: the basal melt parameterisation. These relationships, together with a time-dependent warming derived from ESMs, then lead to a time-dependent mass loss of the ice sheet. This method was applied by Levermann et al. (2014, 2020) to a number of ice sheet models. In those studies, CMIP5 models were used to diagnose the relationship between global surface air temperature (GSAT) and ocean temperature changes around Antarctica, and GSAT was used as a driver of the method. The advantage of using GSAT over ocean temperature changes as a driver is that also uncertainties in GSAT changes were included in the uncertainty estimate. Furthermore, GSAT is easier to derive, but it does not account for (future changes in) Southern Ocean dynamics. It could be expected that a regional metric has a better relation with forcing underneath ice shelves. Therefore, the current study improves this step by using subsurface ocean temperature as the driver (Lambert et al., 2021). In addition to the linear melt parameterisation as in the Levermann et al. (2020) study, a more advanced quadratic basal melt parameterisation is applied since observation-based evidence suggests a nonlinear relationship between melting and ocean temperature (Jenkins et al., 2018).

The basal melt parameterisations are calibrated on the sea level contribution derived from observation-based changes in grounding line ice discharge (Rignot et al., 2019), rather than on basal melt as is done in ISMIP6. One advantage of using ice discharge measurements is that they capture the entire ice sheet through satellite measurements of ice height and velocity and therefore are better constrained than basal melt estimates which are not measured for the full ice sheet and for the full time period that we use for calibration. Moreover, when using basal melt for calibration, basal melt observations are required long before the actual ice discharge acceleration takes place due to the delayed response of ice discharge to basal melt. The advantage of this new approach is that ice discharge acceleration during the historical period is directly derived from observations. Since basal melt has a delayed impact on ice discharge, using ice discharge observations for calibration constrains the basal melt even before the observational period. As a calibration target, the mass loss estimates of Rignot et al. (2019) were chosen over Shepherd et al. (2018) for two reasons. The first reason is that Rignot et al. (2019) does not include surface mass balance processes which makes the data better comparable with the linear response functions that only represent the contribution of Antarctic dynamics. The second reason is that the Rignot et al. (2019) record starts earlier which allows us to look into mass loss acceleration during a longer period. Two different calibration methods are applied: a regional calibration on the Amundsen sector and one at the continental scale. By applying the same melt relation to the past and the future, we ensure that the

physics is consistent with four decades of observed mass loss. Here, the assumption is that no new processes are taking place. Using different warming scenarios and RFs for a variety of models, we arrive at a new estimate of the future ice mass loss of Antarctica and the Amundsen sector that is constrained by observed ice discharge.

## 2   Methodology

In this study the contribution of changes in Antarctica's ice discharge to sea level changes is computed with state-of-the-art
ESMs from Coupled Model Intercomparison Project Phase 6 (CMIP6; Eyring et al. 2016) and linear response functions from the Linear Antarctic Response MIP (LARMIP-2; Levermann et al. 2020) ice sheet models. The basic procedure of this study follows that of Levermann et al. (2020) with a number of modifications. First, we give a brief explanation of the procedure as illustrated in Fig. 1.

All computations are performed for five ocean sectors around the Antarctic continent (Fig. 2). The regional mean subsurface
ocean temperatures are taken from each CMIP6 ESM and bias-adjusted with a global ocean reanalyses dataset (Sect. 2.1). Then basal melt is computed from these bias-adjusted temperatures with a basal melt parameterisation and a first guess for the basal melt sensitivity (calibration parameter) which is derived from basal melt observations (Sect. 2.2). The resulting basal melt anomalies are fed into the linear response functions to compute the regional sea level contribution for each of the five sectors (Sect. 2.3). The sum of the five regions gives the summed Antarctic sea level contribution. The calibration
starts either after the regional sea level computation (regional calibration) or after computing the summed Antarctic response (Antarctic-wide calibration) (Sect. 2.4). For each ESM-RF combination, the resulting sea level contribution is compared with observed grounding line ice discharge from Rignot et al. (2019). The basal melt sensitivity is used as the calibration parameter to improve the fit with observations. This is an iterative procedure. The calibration is performed on the Amundsen region (regional calibration) and for the sum of all regions (Antarctic-wide). Optionally, a model selection could be performed based
on a comparison with observed ice discharge (Rignot et al., 2019). Details of each step are described in the subsections that follow.

### 2.1   Ocean forcing

Earth system models from CMIP6 are used as a basis for the computations, guaranteeing implementation of state-of-the-art models in the analysis and projections. The ocean forcing consists of annual mean simulated subsurface ocean temperatures
which are obtained from ESM output instead of estimating them from scaling coefficients and GSAT as in LARMIP-2 (Lambert et al., 2021). The ocean temperatures are taken from the historical experiment (1850-2014) and the Shared Socioeconomic Pathway (SSP) scenarios SSP1-2.6, SSP2-4.5 and SSP5-8.5 (2015-2100). Only models that have data available at the Earth System Grid Federation (ESGF) data server for the historical experiment and all three SSP scenarios (at the time of study) are considered. In addition, models should provide data for the full period (1850-2100) without any data gaps since the computation
of the delayed ice sheet response to basal melt requires a continuous time series. Table 1 summarises which models have been taken into account.

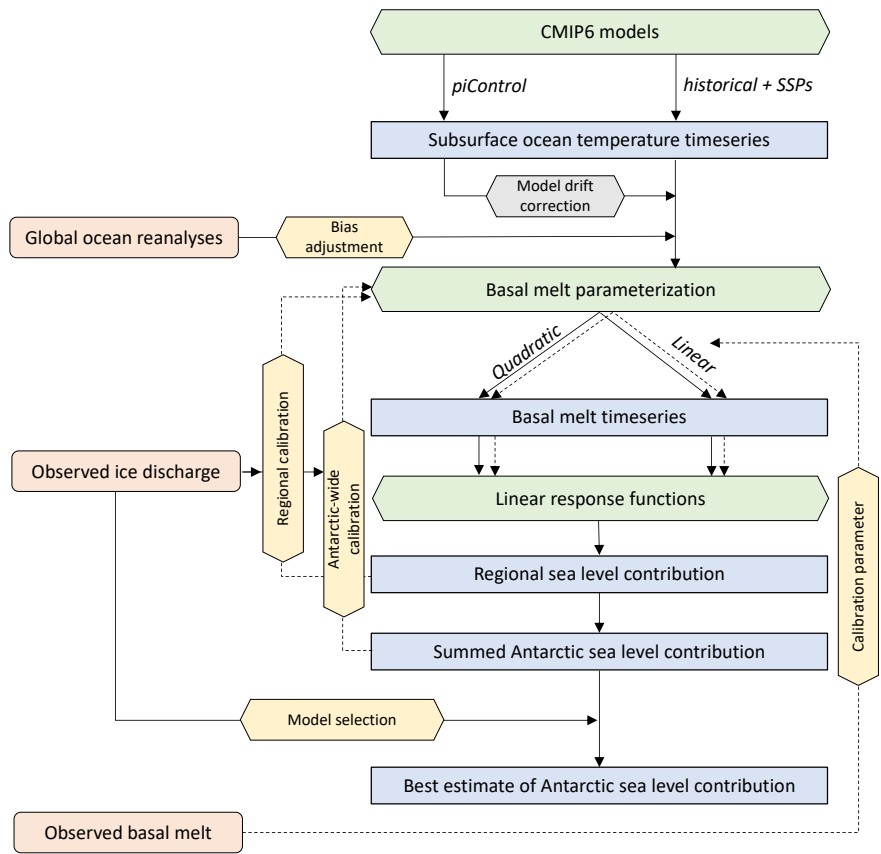

**Figure 1.** Flow diagram of procedure. Observational constraints are indicated in orange, main computations of the Levermann et al. method in green (including model experiments by the modelling groups), calibration methods in yellow, bias-adjustment in grey and (intermediate) output data in blue. The continuous lines represent direct pathways while the dashed lines refer to iterative processes or optional choices during calibration.

Ocean temperatures are averaged over five oceanic sectors: the East AIS (EAIS), Ross, Amundsen, Weddell and Peninsula sector (Fig. 2), and averaged vertically over a range of 100 m, centered around the depth of the ice shelf base (Table 2). In addition, temperatures in an ocean layer around the depth of the continental shelf near the ice shelf front (800-1000 m) were used to assess the impact of thermal forcing depth on the projections (Table 3)(Sect. 3.3.2). The deeper ocean layer is chosen as it approximately represents the deeper water masses on the continental shelf that have access to the cavities under ice shelves. Different from Levermann et al. (2020), the Peninsula sector is defined as a separate ocean sector rather than using the same ocean sector coordinates as the Amundsen sector.

The ocean temperature time series are corrected for model drift by removing the long term trend diagnosed by the linear trend in the pre-industal control (piControl) experiment (Fig. 3). For models that did not provide suitable data for the pi-

**Table 1.** CMIP6 ESMs that have been evaluated. For each region the subsurface ocean temperature bias (in K) compared to the GREP reanalysis is indicated over the period 1993-2018, including years 2015-2018 for the SSP2-4.5 scenario. The 'drift correction' column indicates whether the piControl experiment was used for model drift correction. The bottom rows show the mean and standard deviation ($\sigma$) of the ESM biases and the mean ocean temperature (in °C) and standard deviation of the GREP reanalysis product.

| CMIP6 ESM | EAIS | Weddell | Amundsen | Ross | Peninsula | Drift correction |
|---|---|---|---|---|---|---|
| ACCESS-CM2 | -0.33 | -0.11 | -1.05 | -1.26 | 0.09 | – |
| CAMS-CSM1-0 | 0.24 | -0.05 | 0.22 | -0.94 | 0.39 | piControl |
| CAS-ESM2-0 | 1.43 | 0.79 | 0.20 | -0.18 | 2.18 | – |
| CMCC-ESM2 | 0.31 | -0.23 | 0.51 | -0.10 | 0.58 | piControl |
| CanESM5 | -0.55 | -0.43 | -0.07 | -0.80 | -0.21 | piControl |
| EC-Earth3 | 0.06 | -0.57 | 1.17 | 0.71 | -0.33 | – |
| EC-Earth3-Veg | -0.10 | -0.58 | 0.84 | 0.44 | -0.34 | piControl |
| GFDL-ESM4 | 0.05 | -0.38 | 0.45 | -1.00 | 0.20 | piControl |
| INM-CM4-8 | -0.37 | 0.32 | -0.66 | -0.17 | 0.19 | piControl |
| INM-CM5-0 | -0.74 | -0.24 | -1.16 | -1.11 | -0.16 | piControl |
| MIROC6 | 0.81 | 0.55 | 1.58 | 1.40 | 0.29 | – |
| MPI-ESM1-2-LR | -0.31 | 0.03 | 0.08 | -0.59 | -0.41 | piControl |
| MRI-ESM2-0 | -0.12 | -0.10 | -0.12 | -0.31 | 0.32 | – |
| NorESM2-MM | -0.92 | -0.45 | -0.71 | -0.84 | -0.74 | piControl |
| Bias Mean | -0.04 | -0.10 | 0.09 | -0.34 | 0.15 | - |
| Bias $\sigma$ | 0.59 | 0.40 | 0.78 | 0.74 | 0.67 | - |
| GREP Mean | 0.53 | -0.79 | 1.37 | -0.18 | -0.24 | |
| GREP $\sigma$ | 0.23 | 0.21 | 0.24 | 0.53 | 0.21 | |

**Table 2.** Mean ice shelf depth (in m) for the five sectors in Fig. 2.

| Sector | Depth (m) |
|---|---|
| EAIS | 369 |
| Weddell | 420 |
| Amundsen | 305 |
| Ross | 312 |
| Peninsula | 420 |

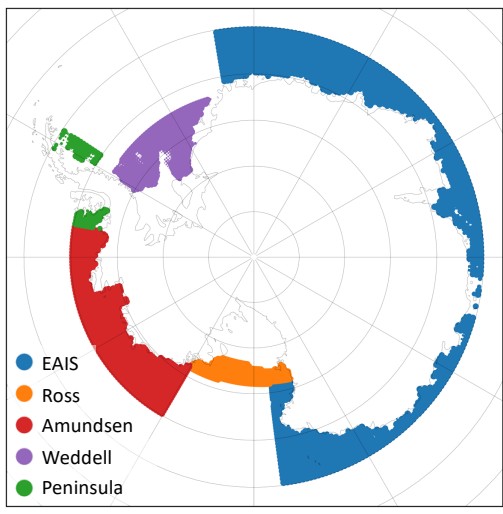

**Figure 2.** Ocean sector definition.

Control experiment, the model drift is not removed. Although the ocean temperature bias has no clear relation with projected temperature trends in ESMs (Little and Urban, 2016), it affects the magnitude of basal melt in the quadratic parameterisation. Therefore, before computing the basal melt the time mean ocean temperatures are bias-adjusted with global ocean reanalyses called the Global Reanalysis Ensemble Product (GREP). GREP can be obtained from the Copernicus Marine Server at 1 de-
gree horizontal resolution over the period during which altimetry data observations are available (1993-2018). It is constructed by postprocessing of four reanalyses: GLORYS2V4 from Mercator Ocean (France), ORAS5 from ECMWF, FOAM/GloSea5 from Met Office (UK), and C-GLORS05 from CMCC (Italy). It should be noted, however, that the reanalysis data may also be biased due to a paucity of assimilated data and the absence of ice shelves in the physical ocean models.

Averaged over all CMIP6 ESMs the subsurface temperature is cold-biased for the EAIS, Weddell and Ross sectors over the
1993-2018 period. For the Amundsen and Peninsula sectors the mean simulated temperature is warm-biased (Table 1). For all regions, the sign of the bias differs between individual models. The ocean temperature time series of the individual models are corrected by the ensemble mean of the reanalysis products over the 1993-2018 time period over the entire historical and future period to obtain the bias-adjusted ocean temperatures (Fig. 3).

## 2.2 Basal melt parameterisation

CMIP6 ESMs do not represent ice shelf cavities and the related thermal and dynamical properties. Coastal ocean temperatures should therefore be translated into these cavities. This can be done by using a parameterisation that relates the far-field (coastal) ocean temperature to melt at the ice shelf base. A large variety of parameterisations exist that link ocean properties to basal melt (Jourdain et al., 2020). Most of the simple basal melt parameterisations assume a relation with thermal forcing, i.e. the

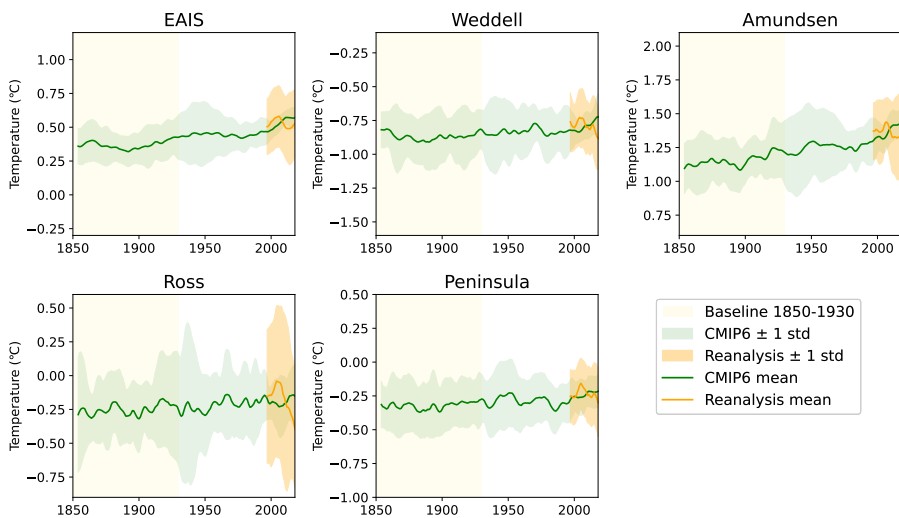

**Figure 3.** Annual mean subsurface ocean temperature time series averaged over all ESMs (green), model drift- and bias-adjusted, and the GREP ensemble mean (orange). Both are smoothed by a five-year running average filter. The temperature is derived from a 100-m thick layer centered around the mean depth of the ice shelf base as specified in Table 2. The historical experiment (1850-2014) is combined with SSP2-4.5 (2015-2018) for this visualisation. Note that the tick distances of the vertical axis are the same for all regions, but the ranges are different.

**Table 3.** Overview of basal melt computation and calibration methods applied in this study. Two different depths were used for the thermal forcing: centered around the mean depth of the ice shelf base (Table 2) and the layer at 800-1000 m depth. Also, two different basal melt parameterisation methods were employed: linear and quadratic. Each parameterisation has been calibrated Antarctic wide and regionally on the Amundsen region. Finally, median basal melt sensitivities used in LARMIP-2 (11.5 m yr K[-1]) and ISMIP6 AntMean method (2.6 m yr K[-2]) have been applied in the linear and quadratic parameterisation, respectively.

| Thermal forcing depth | Parameterisation relation | Basal melt sensitivity |
|---|---|---|
| Ice shelf base | Quadratic | Amundsen calibration |
| 800-1000 m | Linear | Antarctic-wide calibration |
| | | ISMIP6 AntMean Median |
| | | LARMIP-2 Median |

difference between the *in situ* temperature of sea water ($T_o$) and the *in situ* freezing-melting point temperature ($T_f$):

$$TF = T_o - T_f. \tag{1}$$


Our main method employs a quadratic melt relation with thermal forcing (Table 3) as the quadratic relation was suggested to outperform a linear relation (Favier et al., 2019). However, we will also apply a linear relation so that we can compare our

results with the linear relation used in Levermann et al. (2020). The linear relation is defined as:

$$m = \gamma_l \left( \frac{\rho_{sw} c_{po}}{\rho_i L_i} \right) TF, \tag{2}$$

where $m$ is the basal melt and $\gamma_l$ is the linear calibration parameter. It assumes a constant heat exchange, independent on the local stratification and circulation. The quadratic relation, adapted from Favier et al. (2019), is defined as:

$$m = \gamma_q \left( \frac{\rho_{sw} c_{po}}{\rho_i L_i} \right)^2 TF|TF|. \tag{3}$$

where the quadratic calibration parameter is $\gamma_q$. The basal melt sensitivity is defined as $\gamma_l \left( \frac{\rho_{sw} c_{po}}{\rho_i L_i} \right)$ for the linear relation and $\gamma_q \left( \frac{\rho_{sw} c_{po}}{\rho_i L_i} \right)^2$ for the quadratic relation. The quadratic relation assumes that the heat exchange scales with the buoyancy-driven cavity circulation and that this scales linearly with the large-scale temperature gradient. The values of the physical constants $\rho_{sw}$, $c_{po}$, $\rho_i$ and $L_i$ are given in Table 4. The freezing-melting point temperature $T_f$ is computed from the ocean salinity $s_o$ at the thermal forcing depth and the mean depth of the ice shelf base $z_b$:

$$T_f = \lambda_1 s_o + \lambda_2 + \lambda_3 z_b. \tag{4}$$

Favier et al. (2019) take $T_o$ and $T_f$ either as local or as the product of local and the average over the entire ice draft of a given sector. The thermal forcing depth is the depth of the ice shelf base or 800-1000 m (Table 3). In the current study, a purely nonlocal forcing is applied, similar to DeConto and Pollard (2016) and Levermann et al. (2020). This is because the linear response functions are derived from a homogeneous melt perturbation over the entire ice draft and therefore a single basal melt value is required per region for each time step. The values of $T_o$ are computed as averages over the five (far-field) oceanic sectors, around the depth of the ice shelf base (see Table 2) or a deeper layer (800-1000 m depth). Since CMIP6 ESMs do not resolve cavities, the far-field ocean temperature is taken. The underlying assumption is that the ocean temperature remains constant while it is advected into the cavity. The computation of $T_f$ is based on a constant salinity value for each oceanic sector, which is computed from the far-field salinity climatology of the reanalysis data. The resulting values of $T_f$ are approximately -1.6 °C in each sector.

Note that the melt is positive if the ocean temperature exceeds the freezing-melting point temperature and negative (i.e. water is refreezing) otherwise. In the current study, basal melt anomalies are used to compute the sea level contribution. The basal melt anomalies are defined as the difference in basal melt between time $t$ and the baseline time period, 1850-1930. This period was chosen since it is long enough to reduce the impact of natural variability on the baseline but short enough so that it doesn't include the trends due to anthropogenic forcing.

## 2.3 Sea level contribution

Linear response functions (RFs) from LARMIP-2 will be used to compute the cumulative sea level contribution $\Delta S$ (in meters) due to a change in basal melt $\Delta m$ for each of the five sectors (Fig. 2):

$$\Delta S(t) = \int_0^t d\tau \, \Delta m(\tau) \cdot RF(t - \tau). \tag{5}$$

**Table 4.** Physical constants.

| parameter | symbol | value | unit |
|---|---|---|---|
| ice density | $\rho_i$ | 917 | kg m$^{-3}$ |
| sea water density | $\rho_{sw}$ | 1028 | kg m$^{-3}$ |
| specific heat capacity of ocean mixed layer | $c_{po}$ | 3947 | J kg$^{-1}$ K$^{-1}$ |
| latent heat of fusion of ice | $L_i$ | $3.34 \times 10^5$ | J kg$^{-1}$ |
| heat exchange velocity | $\gamma$ | calibrated | m s$^{-1}$ |
| liquidus slope | $\lambda_1$ | -0.0575 | $^\circ$C PSU$^{-1}$ |
| liquidus intercept | $\lambda_2$ | 0.0832 | $^\circ$C |
| liquidus pressure coefficient | $\lambda_3$ | $7.59 \times 10^{-4}$ | $^\circ$C m$^{-1}$ |

The sum of the five regional sea level contributions gives the total Antarctic sea level contribution.

LARMIP-2 provides RFs of 16 ice sheet models. Combined with the 14 ESMs (Table 1), this results in 224 ESM-RF
combinations for the projections.

## 2.4 Calibration

Basal melt parameterisations are usually calibrated on observed melt rates (Jourdain et al., 2020). In contrast, the basal melt
parameterisations in our study are calibrated on observed ice discharge from Rignot et al. (2019) (grey lines in Fig. 6). This is
done for each individual ESM-RF pair. The basal melt parameterisation can be calibrated with the heat exchange velocity $\gamma$.
It should be noted that $\gamma_l$ and $\gamma_q$ have a different order of magnitude in the linear and quadratic parameterisation, respectively,
and are not directly comparable. The root-mean-square error (RMSE) between the observed and modelled cumulative changes
in ice discharge for each year, weighted equally, over the period 1979-2017 for each ESM-RF pair is determined over a wide
range of $\gamma$ values for Eq. (2) and Eq. (3).

$$\text{RMSE} = \sqrt{\frac{\sum_{t=1}^{T}(\Delta S_{\text{simulated}}(t) - \Delta S_{\text{observed}}(t))^2}{T}} \tag{6}$$

The RMSE is computed over the full time series to constrain models on the cumulative sea level contribution as well as
the acceleration. The $\gamma$ value giving the lowest RMSE for each ESM-RF pair provides the calibrated basal melt sensitivity.
Since the observational uncertainty is small compared to the intermodel spread (Fig. 6), it was not taken into account in the
calibration. Note that this calibration step is a key difference with Levermann et al. (2020). That study did not calibrate the
basal melt parameterisation on ice discharge, but used melt sensitivities derived from observations.
The calibration is applied regionally on the Amundsen region and Antarctic-wide (Table 3), resulting in two basal melt
sensitivities for each ESM-RF pair for a given parameterisation. Figure 5 shows the basal melt sensitivities corresponding with
the calibrated $\gamma$ values for the linear and quadratic basal melt parameterisation and for the two calibration regions. For the

Antarctic-wide calibration, the same $\gamma$ value is applied to each region. The smallest RMSE between the summed discharge over all regions in observations and models determines the calibrated $\gamma$ value. For the Amundsen calibration, the calibrated $\gamma$ value is determined by the best fit between the modelled response and observations over only the Amundsen region. The resulting $\gamma$ values are then applied to the other four regions to obtain the Antarctic summed response.

In addition, to assess the impact of our calibration method on the sea level projections, a single basal melt sensitivity (i.e. the calibration parameter $\gamma$) derived from observed basal melt has been applied to all ESM-RF pairs. This parameter is derived from the median basal melt sensitivity that was used in LARMIP-2 for the linear parameterisation and the median value of the mean Antarctic (AntMean, originally MeanAnt) basal melt sensitivity distribution applied in ISMIP6 for the quadratic parameterisation (Table 3). The ISMIP6 AntMean option was chosen over the Pine Island's grounding line (PIGL) option since the AntMean parameterisation performs better in reproducing observed melt rates in the Amundsen region as well as for the total AIS (Jourdain et al., 2020).

For all basal melt computation and calibration methods, the sea level contributions of the Amundsen region and the total AIS are analysed. The RMSE between observed and modelled ice discharge for these two regions was used to assess the impact of model selection on projections of the Antarctic dynamics contribution to sea level.

## 3 Results

### 3.1 Basal melt computation and calibration

Basal melt is computed from subsurface thermal forcing anomalies from CMIP6 ESMs. The subsurface ocean temperature time series near the mean depth of the ice shelf base over the historical period are shown in Figure 3. Figure 4 shows the thermal forcing for part of the historical and future period (1950-2100). Over the 21st century, all regions show a median increase in thermal forcing but the magnitude varies between individual regions and becomes scenario dependent around year 2050.

The basal melt parameterisations are calibrated by fitting the sea level response of each ESM-RF pair on the changes in observed ice discharge over the full 1979-2017 period (Rignot et al., 2019). This exercise shows that the median basal melt sensitivity value resulting in the lowest RMSE differs between the Antarctic-wide calibration and calibration on the Amundsen region (Fig. 5). For the Amundsen region a higher median basal melt sensitivity than for the Antarctic-summed response improves the fit. The Antarctic-wide calibration includes regions with a small or negative past contribution to sea level, resulting in a lower basal melt sensitivity. The relatively high magnitude of the median basal melt sensitivity of the Amundsen region is consistent with the higher sensitivity to ocean warming as described in Dinniman et al. (2016). The contribution of ice discharge to sea level over the observational period is positive and (at least partly) attributable to ocean warming for both the Amundsen region and the total AIS (Pritchard et al., 2012). Therefore, for each ESM-RF pair the calibration parameter, and thus the basal melt sensitivity, should be positive for both Antarctica and the Amundsen region. If the best fit (lowest RMSE) is associated with a negative basal melt sensitivity, this means that the ESM-RF combination could not be calibrated. Between 83% and 90% of all ESM-RF pairs could be calibrated, dependent on the parameterisation type and calibration region, as indicated on top of the boxes in Fig. 5. These percentages show that for the Antarctic-wide calibration region, the quadratic

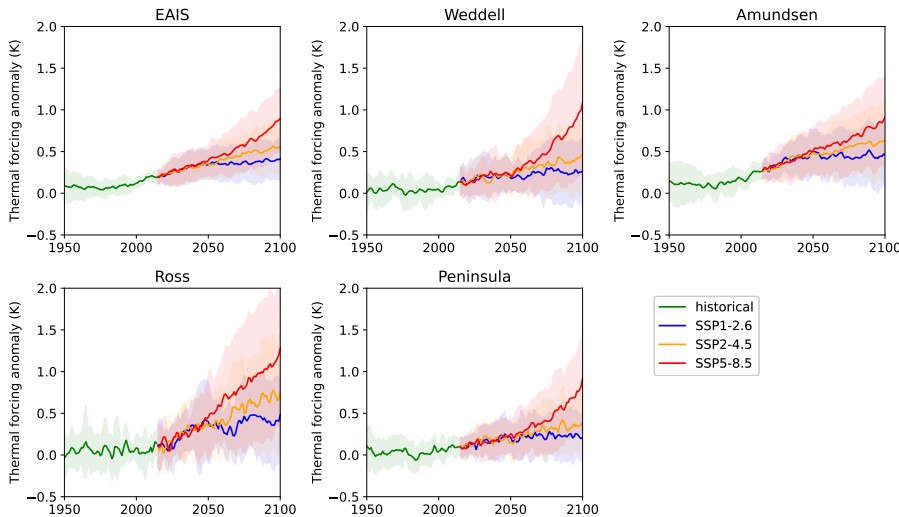

**Figure 4.** Thermal forcing anomalies centered around the mean depth of the ice shelf base for SSP1-2.6, SSP2-4.5 and SSP5-8.5 including all evaluated CMIP6 ESMs (Table 1) from 1950 to 2100 relative to the baseline period 1850-1930. The shaded regions indicate the intermodel spread (17th to 83rd percentiles) in ocean subsurface temperature between the ESMs.

parameterisation has a higher percentage of positive values than the linear parameterisation. The boxplots only represent the ESM-RF pairs with positive basal melt sensitivities. These calibrated ESM-RF pairs are used in the hindcasts and projections of changes in ice discharge.

For the linear parameterisations, we compared our calibrated basal melt sensitivities to the values used in LARMIP-2 (Levermann et al., 2020) (green shading in Fig. 5). This comparison shows that our Antarctic-wide calibration results in a median basal melt sensitivity just below the lower bound of the LARMIP-2 interval. Regional calibration on the Amundsen sector results in a median basal melt sensitivity above the LARMIP-2 range. Furthermore, the spread in our calibrated basal melt sensitivities is much larger than the spread in the observation-based range. For the Amundsen calibration, more than half of the ESM-RF pairs have a higher calibrated basal melt sensitivity than the observation-based LARMIP-2 range. These ESM-RF pairs will underestimate historical ice discharge in the Amundsen region when applying the lower, observation-based melt sensitivity. Vice versa, for the Antarctic-wide calibration, about half of the ESM-RF pairs have a lower calibrated sensitivity than the LARMIP-2 range. These ESM-RF pairs will overestimate historical ice discharge for the total AIS when applying the higher melt sensitivity from within the LARMIP-2 range.

A similar comparison was made for the quadratic parameterisation, with the basal melt sensitivities applied in ISMIP6 (Jourdain et al., 2020). Two different basal melt calibration options were used in ISMIP6: the AntMean method and the Pine Island's grounding line (PIGL) method. Our median Antarctic-wide calibrated basal melt sensitivity sits at the lower end of the range of the ISMIP6 AntMean calibration option (blue shading in Fig. 5), which reproduces basal melt rates around the Antarctic continent. The Amundsen calibration on ice discharge results in a median basal melt sensitivity at the top end of the ISMIP6

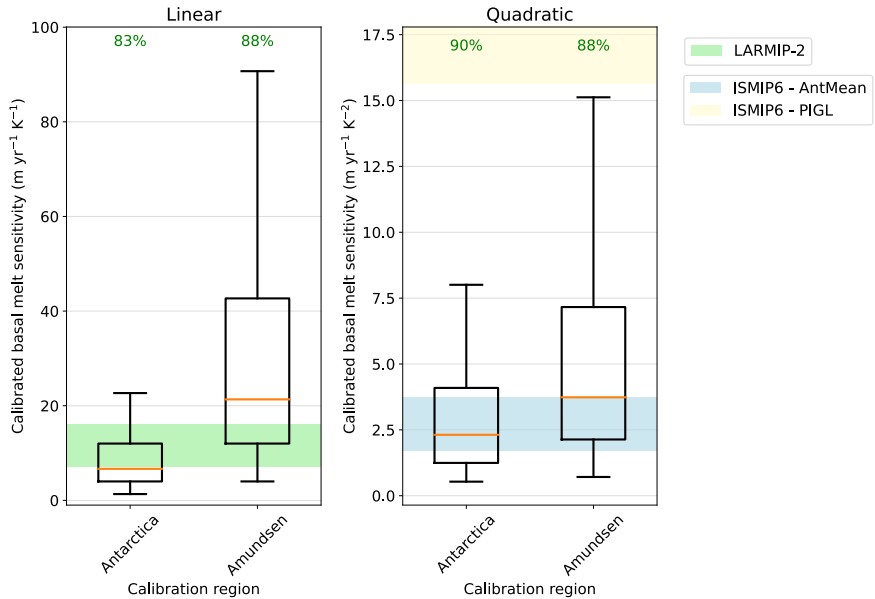

**Figure 5.** Box-and-whisker plots of basal melt sensitivity values corresponding with the calibrated $\gamma$ values of ESM-RF pairs. Only the sensitivities of calibrated $\gamma$ values greater than zero are shown in the plot. The percentage of ESM-RF pairs with positive $\gamma$ values is indicated by the green values on top of the boxes for each region. The horizontal orange line indicates the median value, boxes indicate the 25-75 percentile range and whiskers the 5-95 percentile range. Values beyond this range are not shown. The shaded regions indicate basal melt sensitivity ranges that are used in other studies. The green shading represents the basal melt sensitivity range of 7-16 m yr$^{-1}$ K$^{-1}$ used in Levermann et al. (2020). The blue and yellow shading indicate the 5-95% range of the basal melt sensitivities corresponding with the $\gamma$ values used for the nonlocal quadratic parameterisation in ISMIP6 (Jourdain et al., 2020) for both the Antarctic mean (AntMean) and Pine Island's grounding line (PIGL) calibration option, respectively. For PIGL the 95% bound is 84 m yr$^{-1}$ K$^{-2}$, which is outside the scale of the vertical axis.

AntMean range. Only some calibrations of ESM-RF pairs outside the 95th percentile range resulted in $\gamma$ values within the ISMIP6 PIGL range (yellow shading in Fig. 5), which reproduces the highest observed basal melt of the AIS. However, it should be remarked that the ISMIP6 PIGL calibration also includes negative ocean temperature corrections all around Antarctica that counter-balance the effects of the large $\gamma$ values (Jourdain et al., 2020). Similar to the linear parameterisation, about half of the ESM-RF pairs has a calibrated melt sensitivity higher than the ISMIP6 AntMean range for the Amundsen calibration. These model pairs will underestimate historical ice discharge in the Amundsen region when applying the ISMIP6 AntMean basal melt sensitivity.

For the quadratic parameterisation, the sensitivity of the calibration parameter to the thermal forcing is tested. In this way, the impact of the uncertainty in the reanalysis dataset on the sea level projections is explored. This has been done by adding a positive temperature perturbation to the temperature time series near the ice shelf base of each ESM. The temperature perturbation is equal in size to one standard deviation between the reanalysis products (see the shaded orange regions in Fig 3).

**Table 5.** Sensitivity of calibration parameter of the quadratic parameterisation to thermal forcing. Values indicate median basal melt sensitivity in m yr$^{-1}$ K$^{-2}$ for calibrated $\gamma$ values based on three types of thermal forcing. The Antarctic-wide calibration (QA) and regional Amundsen calibration (QR) are shown. For comparison the median value of the AntMean calibration that is used in ISMIP6 (QM) is shown. The first thermal forcing type is the thermal forcing as shown in Fig. 3, which is based on the bias-adjusted ocean subsurface temperature timeseries of the ESMs near the ice shelf base. The second type is based on the same ocean temperature timeseries raised with one standard deviation ($1\sigma$) that expresses the spread between the ocean reanalysis products (GREP $\sigma$ in Table 1). The third type is the thermal forcing at 800-1000 m depth.

| Thermal forcing | Antarctic-wide (QA) [m yr$^{-1}$ K$^{-2}$] | Amundsen (QR) [m yr$^{-1}$ K$^{-2}$] | ISMIP6 AntMean (QM) [m yr$^{-1}$ K$^{-2}$] |
|---|---|---|---|
| Ice shelf base | 2.3 | 3.7 | 2.6 |
| Ice shelf base + $1\sigma$ | 1.8 | 3.4 | - |
| 800-1000 m | 1.2 | 5.5 | - |

The resulting calibrated basal melt sensitivities are listed in Table 5 (Ice shelf base + $1\sigma$). As expected, the higher ocean temperatures lead to stronger forcing in the quadratic parameterisation and therefore a lower basal melt sensitivity is required for the best fit with observations.

To summarise, a comparison of the calibrated basal melt sensitivity values in our study and equivalents in LARMIP-2 (Levermann et al., 2020) and the ISMIP6 AntMean method (Jourdain et al., 2020) suggests that calibration on past ice discharge rather than on basal melt observations results in relatively low basal melt sensitivities for the Antarctic-wide calibration. The Amundsen sector is more consistent with the high end of the basal melt sensitivity ranges applied in LARMIP-2 and the AntMean calibration option of ISMIP6. It should be noted that calibration on ice discharge leads by definition to a better fit with past ice discharge for individual ESM-RF pairs. Remarkably, the spread in the calibrated melt sensitivities is much higher than the observation-based ranges of LARMIP-2 and the ISMIP6 AntMean method. ESM-RF pairs with calibrated melt sensitivity values outside the observation-based ranges either underestimate or overestimate past ice discharge when using observation-based sensitivities.

## 3.2 Hindcasts of Antarctic and Amundsen sea level contribution

Hindcasts of the dynamic contribution of the Amundsen region and the total AIS to sea level rise are made to assess how well changes in ice discharge could be reproduced after calibration over the period 1979-2017. The calibration is performed by fitting the sea level on observations using a least squares fit of the sea level contribution for each year, weighted equally, over the hindcast period. The results of the linear and quadratic parameterisation are about equal when applied to the region of calibration (same RMSE; Table 6). However, the quadratic parameterisation performs better (lower RMSE) after calibration on an independent region than the linear parameterisation (i.e. when calibrated on the Amundsen region and applied to the total

**Table 6.** RMSEs of the least squares fit of the median sea level contribution of each year, weighted equally, between calibrated results and ice discharge observations of Rignot et al. (2019). Results are shown for combinations of the two parameterisations, linear and quadratic (Q) and two calibration methods, regional Amundsen (R) and Antarctic-wide (A), for two hindcast regions: Antarctic Ice Sheet (AIS) and the Amundsen region.

| Basal melt method | RMSE AIS [mm] | RMSE Amundsen [mm] |
|---|---|---|
| Linear Amundsen | 14.9 | 1.4 |
| Quadratic Amundsen (QR) | 7.2 | 1.4 |
| Linear Antarctic-wide | 1.7 | 2.7 |
| Quadratic Antarctic-wide (QA) | 1.6 | 2.4 |

AIS or vice versa). Observations confirm that the quadratic relation can be better used when calibrating on (partly) independent regions (Jenkins et al., 2018). In the remainder of this article, therefore, our main results are based on the quadratic basal melt parameterisation. The linear parameterisation is used for making projections with the LARMIP2 median basal melt sensitivity (Sect. 3.3.1). The differences in the projections between the quadratic and linear parameterisation are further discussed in Sect. 3.3.2.

Figure 6 shows the hindcasts of all ESM-RF pairs using the calibrated basal melt sensitivities (Fig. 5). The two panels show the hindcasts for the total AIS and the Amundsen region, as specified in the titles. The total Antarctic sea level response is based on the summed contribution over the five sectors (Fig. 2). The colors represent two calibration methods, where red is the calibration on the Amundsen region and blue the Antarctic-wide calibration. The observed ice discharge values (Rignot et al., 2019) are shown in grey.

First, we evaluate the cumulative magnitude of the modelled sea level contributions over the period 1979-2017 (Table 7). The median Amundsen calibration overestimates the cumulative AIS contribution by about 30% whereas the median Antarctic-wide calibration underestimates the contribution by about 10%. For the Amundsen region, the cumulative contribution is underestimated by the median response of the Amundsen calibration (ca. 20%) and strongly underestimated by the Antarctic-wide calibration (ca. 60%). Both calibration methods do not give an agreement in terms of the cumulative sea level contribution because of the choice to calibrate on the time series rather than on the cumulative sum. Even though the Antarctic-wide calibration is (by construction) closer to the observed Antarctic ice discharge than the Amundsen calibration, the strong underestimation of the Amundsen region still means that the response in other regions is overestimated. It should be kept in mind that the errors in the individual regions compensate each other, resulting in a summed Antarctic response that is close to observations.

Second, we evaluate the evolution of the sea level response over time. For the Antarctic-wide calibration, the median value overestimates changes in Antarctic discharge before 2001 and underestimates them thereafter. This means that the sea level acceleration over the full period cannot be captured with the Antarctic-wide calibration, making it likely that it will be underes-

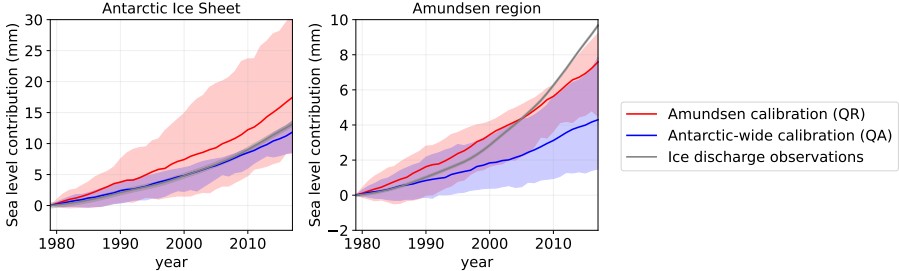

**Figure 6.** Impact of calibration target region on sea level illustrated by hindcasts showing the sea level contribution over the period 1979-2017 based on all calibrated ESM-RF pairs for the total AIS (left panel) and the Amundsen region (right panel). The historical experiment is extended with SSP2-4.5 scenario for the years 2015-2017. The red lines indicate the median contribution based on the regional Amundsen calibration, whereas the blue lines indicate the median contribution for the Antarctic-wide calibration. Only the quadratic parameterisation with thermal forcing near the ice shelf base is shown. The observation-based changes in ice discharge from Rignot et al. (2019) are shown in grey. The shaded area indicates the associated likely range (17th to 83rd percentiles) for the modelled response and the observational error for the Rignot et al. (2019) data.

timated in future projections as well. This is also visible in the ice discharge rate over the last decade of the hindcast (Table 7), which is lower than in observations. In a similar way, the Amundsen calibration overestimates the changes in Amundsen discharge before 2005 and underestimates them thereafter. So for the Amundsen region, even when using the Amundsen-specific calibration, the acceleration is not captured by the median response and the rate over the last decade of the hindcast is underes-

timated. This means that despite its overestimation of the cumulative sum over the hindcast period for the AIS, the Amundsen calibration will presumably underestimate future projections of the sea level contribution for the Amundsen region. It should be noted that not just the acceleration of the Amundsen contribution cannot be reproduced, but the relative dominance of Amundsen with respect to the total Antarctic contribution cannot be reproduced either (about 70% in observations, about 30-40% in our results).

Since the Amundsen region is the most important contributing region to the summed Antarctic response over the hindcasting period, we tested whether a selection of models could better capture past ice discharge in the Amundsen region. The top 10% calibrated models with the best fit to ice discharge observations (Fig. A1) were selected for both the Amundsen and Antarctic-wide calibration. The selection was based on the model performance in the calibration region. As a logical consequence, the top 10% ESM-RF pairs from the two calibration methods performs better on the cumulative sea level contribution in

the calibration region (Table 7). Interestingly, the same selection of models also performs better in the region that was not used for the calibration. After Antarctic-wide selection the Amundsen sea level contribution in the hindcasts are closer to observations. Unfortunately, for the Antarctic-wide selection, the contributions of the other regions increase as well, which increases their error relative to observations. The Amundsen selection resulted in higher estimates than for the full model suite in the Amundsen region itself (by construction), but lower estimates (closer to observations) for the Antarctic summed

**Table 7.** The median cumulative sea level contribution ($\Delta S$) over the hindcast period 1979-2017 and the rate (d$S$/d$t$) over the last decade (2008-2017) of the hindcast period for the two calibration methods (Amundsen and Antarctic-wide) and for the ice discharge observations of Rignot et al. (2019). Results are shown for the quadratic basal melt parameterisation with thermal forcing near the ice shelf base.

| | AIS | | Amundsen region | |
|---|---|---|---|---|
| Source | $\Delta S$ [mm] | d$S$/d$t$ [mm yr$^{-1}$] | $\Delta S$ [mm] | d$S$/d$t$ [mm yr$^{-1}$] |
| Ice discharge observations | 13.1 | 0.58 | 9.7 | 0.48 |
| Amundsen calibration (QR) | 17.5 | 0.84 | 7.6 | 0.27 |
| Antarctic-wide calibration (QA) | 11.8 | 0.45 | 4.3 | 0.17 |
| Amundsen calibration (QR) - top 10% | 16.6 | 0.86 | 9.3 | 0.44 |
| Antarctic-wide calibration (QA) - top 10% | 13.3 | 0.60 | 5.3 | 0.24 |

response. As a result, the Amundsen contribution relative to the total AIS improves after model selection on the Amundsen region.

## 3.3 Sea level contribution projections

In this section, projections of the sea level contribution due to basal melt for the AIS and the Amundsen region are presented. The projections comprise the 21st century. Computations start in the year 1850 so that the delayed contribution of ice discharge 335 due to basal melt is included in the future sea level response. Figure 7 shows our main projections for the SSP5-8.5 scenario, based on the calibrated basal melt sensitivities for the quadratic parameterisation and thermal forcing near the ice shelf base. We assess two metrics: the cumulative magnitude and the rate of the sea level response. The cumulative sea level response is computed by taking the difference between the year 2100 and the average over the period 1995-2014. The sea level response rate at the end of the 21st century is indicative of differences in committed sea level rise beyond 2100. The sea level response 340 rate is computed by a linear regression on the sea level response over the period 2081-2100.

First, we present the calibrated projections for the three SSP scenarios and explore the impact of calibration on projections of the sea level contribution. Second, the sensitivity of projections to methodological choices, such as the parameterisation relation (quadratic/linear), thermal forcing depth (ice shelf base/800-1000 m) and model selection (Earth system model/Ice sheet model) is explored.

### 345 3.3.1 Impact of calibration on sea level projections

To understand how calibration of individual ESM-RF combinations on past ice discharge influences the results compared to using observation-based basal melt sensitivities, we also made projections in which a single basal melt sensitivity is applied

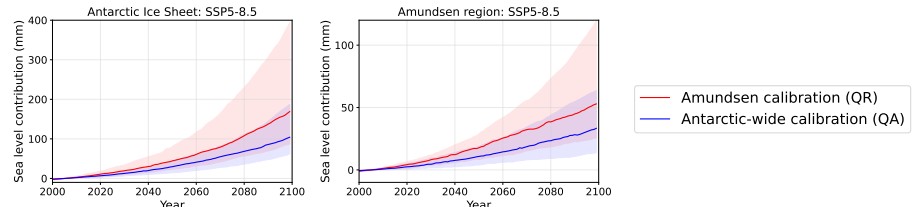

**Figure 7.** Projections showing the calibrated sea level contribution over the period 2000-2100 based on SSP5-8.5, for the total AIS (left panel) and the Amundsen region (right panel). The red lines indicate the median contribution based on the regional Amundsen calibration, whereas the blue lines indicate the median contribution for the Antarctic-wide calibration. Results are shown for the quadratic parameterisation and thermal forcing near the ice shelf base. The shaded area indicates the associated likely ranges (17th to 83rd percentiles).

in all ESM-RF combinations. This single value is the median basal melt sensitivity applied in LARMIP-2 (11.5 m yr K$^{-1}$) (Levermann et al., 2020) for the linear parameterisation (LM) and the median nonlocal basal melt sensitivity applied in ISMIP6

for the AntMean method (2.6 m yr K$^{-2}$) (Jourdain et al., 2020) for the quadratic basal melt parameterisation (QM). The resulting projections from these basal melt computation methods are included in Figures 8 and 9 (Median MIP $\gamma$). In these figures, the green numbers correspond with the median values of the projections. The median projected values are used to quantify the impact of the basal melt method on the sea level projections.

First, the sea level contribution of the total AIS is analysed. Figure 8 shows the projected sea level response for each SSP sce-

nario and different basal melt computation methods. The computation methods include the median MIP basal melt sensitivities (QM, LM) and the calibrated sensitivities (QA and QR). The top panels represent the cumulative projections and the bottom panels the sea level response rates over the period 2081-2100. Not surprisingly, a higher emission scenario leads to a higher sea level contribution. Absolute differences between the basal melt computation methods within one SSP scenario become more explicit for the higher emission scenarios, but relative differences within one SSP scenario are comparable. To compare relative

differences we use the ratio of the sea level projections between the highest and lowest basal melt method, which is QR/QA for the AIS sea level contribution. The ratio QR/QA (1.6) is only slightly larger than the ratio between the SSP5-8.5 and SSP1-2.6 scenario (1.4; averaged over all methods), indicating that the influence of the basal melt computation method on the sea level response is more or less similar to the impact of the emission scenarios. Since the highest sea level projections result from the Amundsen calibration method and the lowest sea level projections from the Antarctic-wide calibration method, this means that

this difference can be entirely attributed to the calibration region.

The projections of the AIS using the median basal melt sensitivities applied in ISMIP6 AntMean (QM) and LARMIP-2 (LM) fall in between the two calibrated projections. This is consistent with the median basal melt sensitivity of LARMIP-2 and IS-MIP6 AntMean, which is located above the median Antarctic-wide calibrated value and below the Amundsen-calibrated value, respectively (Fig. 5). Even though the spread between the basal melt methods is extended by using the calibration methods, us-

ing single basal melt sensitivities based on basal melt observations with different parameterisation types (linear/quadratic) also leads to a large spread in the projections (LM/QM = 1.3; averaged over all SSPs). We remark that the top 10% best-performing

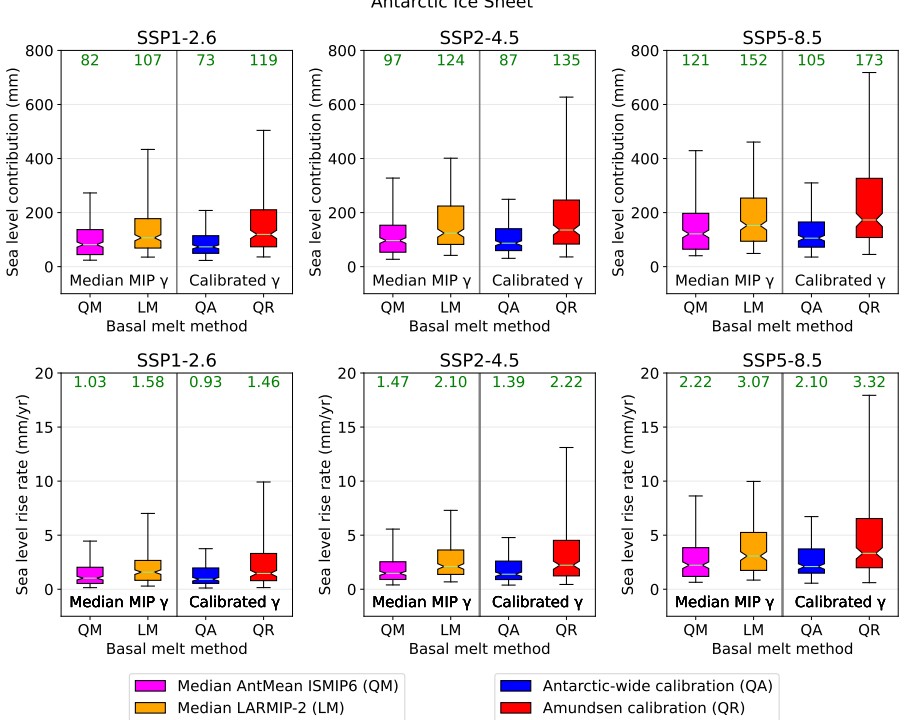

**Figure 8.** Projected Antarctic sea level response for SSP1-2.6, SSP2-4.5 and SSP5-8.5. Top panels show the sea level contribution in 2100 compared to the period 1995-2014 and bottom panels the sea level rise rates over the period 2081-2100. The spread is determined by the (calibrated) ESM-RF pairs. The green numbers indicate the median values (corresponding with the green lines), whereas the boxes show the 25-75 percentiles and the whiskers the 5-95 percentiles. The left hand side shows projections using a single median basal melt sensitivity from the ISMIP6 AntMean method (QM) and from LARMIP-2 (LM). The basal melt computation methods on the right hand side are our main projections with calibrated basal melt sensitivities on ice discharge observations of the Amundsen region (QR) and the total AIS (QA). ESM-RF pairs that could not be calibrated are removed from all basal melt methods so that the same models are included in the comparison. If ESMs did not simulate year 2100, 2099 was used instead.

models in reproducing ice discharge observations (Fig. A1), result in estimates that fall in between the Antarctic-wide (QA) and Amundsen calibration (QR) methods, reducing the spread (Fig. A2).

As a next step, the AIS sea level response rates are assessed at the end of the 21st century (2081-2100). These are important for sea level differences beyond 2100. The ratio QR/QA for the sea level response rates (1.6; averaged over all SSPs) shows that the influence of the calibration region on the response rate is smaller than the effect of the SSP scenarios (SSP5-8.5/SSP1-2.6 = 2.1; averaged over all basal melt methods). The effect of the SSP scenarios is stronger for the quadratic parameterisations (QM, QA, QR) than for the linear one (LM). Consequently, the highest median response rate in SSP5-8.5 and SSP2-4.5 is using the QR basal melt method, whereas in SSP1-2.6 the response rate based on the median LARMIP-2 basal melt sensitivity (LM) is highest. This could be explained by the linear (rather than quadratic) relation with thermal forcing (see Sect. 3.3.2),

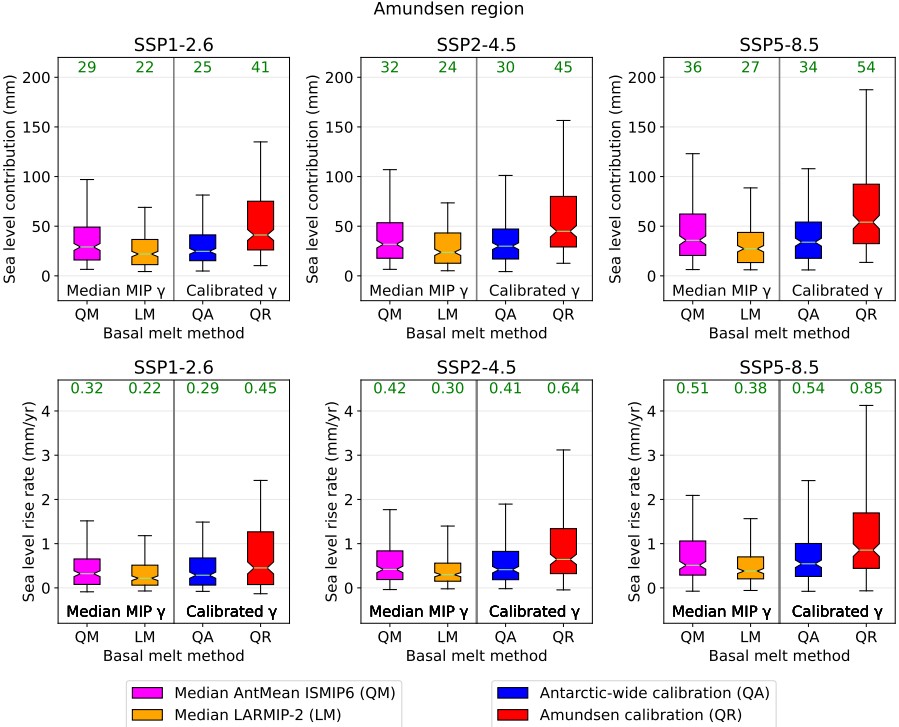

**Figure 9.** Same as Fig. 8 but for the Amundsen region.

which is independent on the absolute ocean temperature (which is linked to the SSP scenarios). It should also be noted that the Amundsen calibration is more skewed towards higher sea level response rates than the other basal melt methods. This is a result of the higher basal melt sensitivities that were required to fit the modelled historical Amundsen sea level contribution to ice discharge observations.

Second, the sea level projections of the Amundsen region are analysed (Fig. 9). For the Amundsen region, the highest projection is given by the Amundsen calibration, whereas the lowest projection is based on the median LARMIP-2 basal melt method. This is a consequence of the linear parameterisation, which is independent on the absolute sea water temperature (Sect. 3.3.2). The ratio of the highest to lowest basal melt method (QR/LM = 1.9) is larger than the ratio between the SSP5-8.5 and SSP1-2.6 scenario (1.3; averaged over all methods), indicating that the influence of the basal melt computation method

on the sea level response is larger than the impact of the SSP scenarios. Also for the Amundsen sea level response rates, the impact of the basal melt method (QR/LM = 2.1) is slightly larger than the impact of the SSP scenario (SSP5-8.5/SSP1-2.6 = 1.8). This demonstrates that the rate is much more sensitive to the SSP scenario than the cumulative sum, indicating increasing differences between SSP scenarios beyond 2100.

     The Amundsen calibration is considered to give the most realistic estimate for future projections of ice discharge in the

Amundsen region. Considering the strong underestimation of past ice discharge rate in the Amundsen region using the

Antarctic-wide calibration (Table 7), we expect that the future projections for the Amundsen region will be too low when using this method. The Amundsen projections using the median LARMIP-2 basal melt sensitivity (LM) are lower than for the Antarctic-wide calibration method and therefore are also expected to underestimate the sea level contribution of the Amundsen region. The projection based on the median ISMIP6 AntMean sensitivity (QM) is probably also too low, since even the hindcasts based on the Amundsen calibration slightly underestimated observed ice discharge in the Amundsen region (Table 7).

We conclude that for the AIS the cumulative sea level variations associated with basal melt computation methods are about equal to variations between different SSP scenarios. For the Antarctic sea level response rate, the SSP scenario is more important than the basal melt method. In contrast, for the Amundsen region the basal melt method impacts the projections (cumulative sum and rate) more than the SSP scenarios. For the Amundsen region, we also conclude that the Amundsen calibration probably gives the most reliable projections since the Amundsen calibration already underestimated past ice discharge and its acceleration in the hindcasts, and the other methods give even lower estimates.

Furthermore, we compared our estimates with the emulated ISMIP6 and LARMIP-2 studies as presented in IPCC AR6 (Table 8). Despite the different method applied, the resulting projections of Antarctica's sea level contribution with the Amundsen calibration are in line with previous multi-model studies. The Amundsen calibration results in median estimates of 0.17 m for SSP5-8.5, 0.14 m for SSP2-4.5 and 0.12 m for SSP1-2.6, sitting in between the emulated ISMIP6 and LARMIP-2 projections, as presented in IPCC AR6 (Fox-Kemper et al., 2021, their Tab. 9.3). It should be noted that this position can only partly be attributed to the calibration on ice discharge observations, since our projections using the median LARMIP-2 sensitivity (LM) result in lower estimates than for LARMIP-2 AR6, which could be attributed to methodological differences other than the basal melt sensitivity. Furthermore, it should be noted that our results using the median ISMIP6 AntMean sensitivity (QM) give lower estimates than emulated ISMIP6, which could be partly explained by a difference in the basal melt sensitivity, since the ISMIP6 emulator samples, apart from the AntMean calibration, also high basal melt sensitivity values from the PIGL calibration. Therefore, the median basal melt sensitivity will be higher in emulated ISMIP6 than the median of the AntMean method, which explains partly the higher projections in emulated ISMIP6 compared to QM. The differences with ISMIP6 and LARMIP-2 will be further discussed in Sect. 4.

### 3.3.2 Impact of methodological choices on projections

In this section we explore what the impact is of several methodological choices on the sea level response projections of the AIS and Amundsen region. These choices include the parameterisation relation (quadratic/linear), thermal forcing depth (ice shelf base/800-1000 m) and model selection (Earth system model/Ice sheet model). Additionally, we further motivate our choices to use the quadratic parameterisation with thermal forcing near the ice shelf base in our main projections (Fig. 7; QA and QR in Figs. 8 and 9).

First we assess the impact of the parameterisation type on the calibrated projections for the AIS and the Amundsen region (Fig. 10). To this end we applied two different parameterisations: a linear (Eq. 2) and a quadratic relation (Eq. 3) with thermal forcing. Both relations are calibrated on observed ice discharge (Fig. 5) using the Antarctic-wide and the Amundsen calibration. The results show that if the parameterisation is used to make projections for the same region as the region that is used for

**Table 8.** Projected dynamic contributions to sea level in meters from the AIS in 2100 relative to 1995-2014. The numbers for LARMIP-2, emulated ISMIP6 and surface mass balance (SMB) contribution are obtained from the IPCC AR6 report (Fox-Kemper et al., 2021, their Tab. 9.3). Note that in our table 'Emulated ISMIP6 AR6 - excl. SMB' represents the 'Emulated ISMIP6 total' minus the SMB contributions (estimated from the AR5 parametric Antarctic Ice Sheet SMB model) to allow comparison with our results, since our study only accounts for the dynamic response. The emulated ISMIP6 number also includes the estimated historical dynamic response. The contribution of LARMIP-2 includes all ice sheet models and the historical dynamic response is incorporated in the method. The columns show the 17th, 50th and 83rd percentiles of the distribution.

| Scenario | Forcing/Source | 17% | 50% | 83% |
|---|---|---|---|---|
| SSP5-8.5/RCP8.5 | Antarctic-wide calibration (QA) | 0.06 | 0.11 | 0.19 |
| | Amundsen calibration (QR) | 0.09 | 0.17 | 0.41 |
| | Median ISMIP6 AntMean sensitivity (QM) | 0.05 | 0.12 | 0.27 |
| | Median LARMIP-2 sensitivity (LM) | 0.08 | 0.15 | 0.32 |
| | Emulated ISMIP6 AR6 - excl. SMB | 0.10 | 0.13 | 0.17 |
| | LARMIP-2 AR6 - excl. SMB | 0.10 | 0.20 | 0.39 |
| SSP2-4.5/RCP4.5 | Antarctic-wide calibration (QA) | 0.05 | 0.09 | 0.16 |
| | Amundsen calibration (QR) | 0.07 | 0.14 | 0.34 |
| | Median ISMIP6 AntMean sensitivity (QM) | 0.04 | 0.10 | 0.22 |
| | Median LARMIP-2 sensitivity (LM) | 0.06 | 0.12 | 0.26 |
| | Emulated ISMIP6 AR6 - excl. SMB | 0.07 | 0.12 | 0.16 |
| | LARMIP-2 AR6 - excl. SMB | 0.09 | 0.17 | 0.33 |
| SSP1-2.6/RCP2.6 | Antarctic-wide calibration (QA) | 0.04 | 0.07 | 0.14 |
| | Amundsen calibration (QR) | 0.06 | 0.12 | 0.28 |
| | Median ISMIP6 AntMean sensitivity (QM) | 0.04 | 0.08 | 0.19 |
| | Median LARMIP-2 sensitivity (LM) | 0.06 | 0.11 | 0.23 |
| | Emulated ISMIP6 AR6 - excl. SMB | 0.06 | 0.11 | 0.15 |
| | LARMIP-2 AR6 - excl. SMB | 0.08 | 0.15 | 0.29 |

calibration, the cumulative sea level contribution is almost equal for both parameterisations. This means that calibration on past ice discharge strongly constrains the future response if applied to the region of projections.

On the other hand, if the calibration is performed in the Amundsen region and applied to make Antarctic projections, or vice versa, clear differences between the linear and quadratic relation appear. For the Amundsen calibration, the quadratic parameterisation results in lower projections for the Antarctic-wide contribution than the linear parameterisation. This can be expected, since the quadratic parameterisation is dependent on the absolute ocean temperature, whereas the linear parameterisation only uses temperature anomalies. By its definition the quadratic relation with thermal forcing implies that sectors that

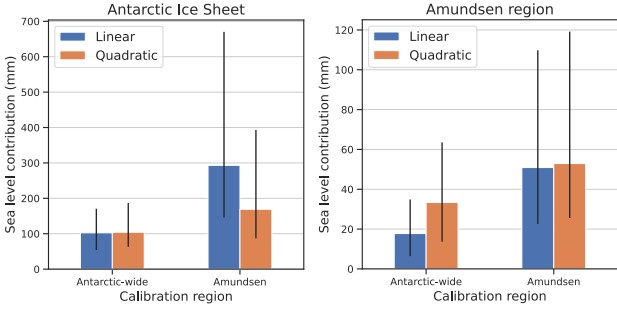

**Figure 10.** Projections of Antarctic sea level contribution for SSP5-8.5 for all calibrated ESM-RF combinations for the AIS (left) and Amundsen region (right). Results are shown for thermal forcing near the ice shelf base. The bars show the median projections for the Antarctic-wide and regional Amundsen calibration using the linear (blue) and quadratic (orange) parameterisations. The spread indicates the 17th to 83rd percentiles.

are melted by warmer waters are more sensitive than the colder sectors, even if the same basal melt sensitivity is applied. So if the Amundsen calibration is applied to colder ocean sectors than the Amundsen sector, this leads to less basal melt for a similar temperature increase, since the ocean temperatures are lower. In a similar way, Antarctic-wide calibration of the linear parameterisation leads to a lower basal melt sensitivity and thus lower projections for the Amundsen region than the quadratic parameterisation.

Favier et al. (2019) demonstrate that the quadratic parameterisation gives better results in representing ocean-induced melting under ice shelves than the linear forcing when compared with ocean–ice-sheet coupled simulations. Furthermore, Holland et al. (2008) show with an ocean model that total ice shelf basal melt increases quadratically as the ocean offshore of the ice front warms. Moreover, the quadratic relationship between thermal forcing and basal melt is confirmed by observations (Jenkins et al., 2018). These arguments are an important motivation to apply the quadratic parameterisation in our study.

Second, we assessed the impact of the thermal forcing depth on the calibrated projections (Fig. 11). For this experiment, thermal forcing and basal melt sensitivity are based on ocean temperature at two different depths: 100 m centered around the mean depth of the ice shelf base (similar to LARMIP-2) and an ocean layer around the depth of the continental shelf near the ice shelf front. The deeper ocean layer is chosen for comparison since the relevant water masses that drive the melting close to the grounding line originate from the deepest depth of the bed near the ice shelf front, which we approximate as 800-1000 m. We only use the quadratic parameterisation, which is dependent on the absolute ocean temperature. Surprisingly, for the deeper layer, the Antarctic-wide calibration leads to a lower basal melt sensitivity, whereas the Amundsen calibration leads to a higher basal melt sensitivity than the corresponding basal melt sensitivities near the ice shelf base (Table 5). This can be explained by the differences in the water temperature and the warming rates of the two layers. For the Amundsen region, the ocean temperature in the deeper 800-1000 m layer warms slower than the ocean temperature near the ice shelf base (Fig. A3), although the temperature itself is comparable in magnitude. Therefore, a higher basal melt sensitivity is required to match ice discharge observations. In contrast, for all other regions, the ocean layer at 800-1000 m depth is warmer than the temperature

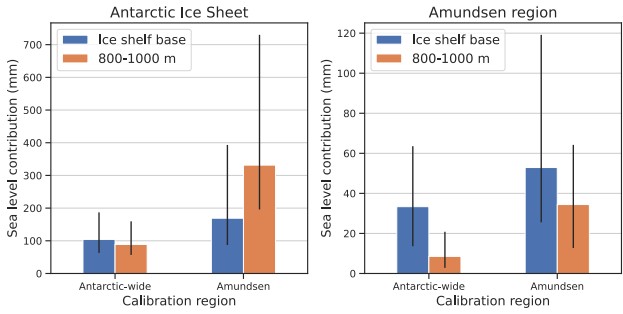

**Figure 11.** Projections of the sea level contribution of the AIS (left) and Amundsen region (right) for SSP5-8.5 for all calibrated ESM-RF combinations using the *quadratic* parameterisation. The bars indicate the median sea level contribution in 2100 relative to 1995-2014. The thermal forcing and basal melt sensitivity are based on ocean temperature at two different depths: 100 m centered around the mean depth of the ice shelf base (blue) and 800-1000 m depth (orange). The black lines indicate the 17th to 83rd percentiles.

near the depth of the ice shelf base, resulting in a higher ocean forcing. In the Weddell, Ross and the Peninsula regions, the
temperature also warms faster in the deeper layer than in the layer at the depth of the ice shelf base, resulting also in stronger ocean forcing. Due to the stronger ocean forcing in the 800-1000 m depth layer, the calibrated basal melt sensitivity is lower for the Antarctic-wide calibration.

For the AIS projections, the lower Antarctic-wide basal melt sensitivity for 800-1000 m depth is largely compensated by a larger ocean forcing for the Antarctic-wide calibration. This results in a similar sea level contribution for the 800-1000 m-based
projections compared to using the thermal forcing near the depth of the ice shelf base. However, the high Amundsen basal melt sensitivity for the 800-1000 m depth combined with the larger Antarctic-wide ocean forcing leads to higher estimates for the AIS projections. Projections for the Amundsen region are oppositely affected. The ocean forcing is smaller at 800-1000 m depth than near the ice shelf base, and combined with a lower basal melt sensitivity for the Antarctic-wide calibration this leads to much smaller projections. For the Amundsen region itself, the higher basal melt sensitivity only partly compensates
for the smaller ocean forcing, resulting in a smaller sea level projection for the forcing at 800-1000 m compared to forcing near the ice shelf base. As a result, the fraction of Amundsen compared to the total Antarctic contribution is larger for the thermal forcing near the ice shelf base than for the 800-1000 m depth layer. Since this fraction was already smaller than in observations in the hindcast experiments using thermal forcing near the ice shelf base (Sect. 3.2), we argue that using thermal forcing near the ice shelf base leads to more realistic results than thermal forcing in the 800-1000 m depth layer.

We conclude that the depth of thermal forcing has a large influence on the resulting sea level contribution in future projections. Most straightforward, it influences the thermal forcing in the projections, which is depth-dependent, but also region-dependent. However, when calibration is applied, the thermal forcing depth also affects the strength of the basal melt sensitivity through its evolution over the historical period. The thermal forcing near the ice shelf base leads to a more realistic contribution of the Amundsen region compared to the total AIS, and is therefore applied throughout this study.

### 3.3.3 Modelling uncertainties associated with Earth System and Ice Sheet Models


In this section, we assess the role of CMIP6 ESMs and RFs of the LARMIP-2 ice sheet models in projection uncertainties for the AIS by comparing the sea level contributions for the Amundsen calibration, which is considered to perform better than the Antarctic-wide calibration for the Amundsen region (Sect. 3.2) and arguably also for the total AIS contribution (Sect. 4). These models cause the spread of the projections for a specific basal melt method (see the shaded regions in Fig. 7 and the error bars in Figs. 8-11). Fig. 12 shows the projected Antarctic sea level contribution for each individual CMIP6 ESM for the Amundsen calibration. Here, the spread for each ESM is determined by the linear response functions of the ice sheet models. Noticeably, the differences between the scenarios are small compared to the differences between individual ESMs, despite the bias adjustment with ocean reanalysis data. As a measure of ESM spread, we compute the standard deviation between the median values (bar heights). The intermodel standard deviation varies from 144 mm for SSP1-2.6 to 205 mm for SSP5-8.5.



The ESM with the strongest median sea level contribution (CAS-ESM2-0) also exhibits the largest warming over the 21st century for each individual ocean sector and has the second highest median calibrated basal melt sensitivity for the Amundsen region (not shown). Also, it has the fourth lowest ranking in reproducing historical ice discharge compared to the other ESMs. Remarkably, the five ESMs with the highest RMSE for the Amundsen region (when comparing their historical performance to ice discharge observations) are amongst the six models with the highest cumulative sea level contribution for the AIS in the projections. This suggests that applying ESM selection based on the performance of ESMs in reproducing ice discharge observations in the Amundsen region would result in lower estimates of the Antarctic dynamics contribution to sea level projections. However, a potential selection of CMIP6 ESMs based on ice discharge can only be considered if the sensitivity of ice discharge to basal melt perturbations is well represented by the linear response functions (Sect. 4).


Fig. 12 also shows the projected Antarctic sea level contribution for the RF of each individual ice sheet model. Here, the spread in the error bars is determined by the CMIP6 ESMs. Similar as for the ESMs, we computed the intermodel standard deviation between ice sheet models as a measure of ice sheet model spread. The standard deviation between the median values varies from 46 mm for SSP1-2.6 to 62 mm for SSP5-8.5.


The RF of the ice sheet model giving the smallest median sea level contribution (GRIS LSC) has the second lowest calibrated basal melt sensitivity for the Amundsen region and could not be calibrated in combination with half of the ESMs. We remark that this RF also gave the smallest signal in LARMIP-2 (Levermann et al., 2020). The RF of the ice sheet model with the smallest calibrated basal melt sensitivity (PISM DMI) also could not be calibrated when combined with the forcing for 6 out of the 14 ESMs. Moreover, GRIS LSC and PISM DMI have the highest RMSE when compared with observed ice discharge. This suggests that RF selection based on reproducing historical ice discharge would result in higher future estimates of the sea level contribution.



We also compared the spread associated with the ESMs and RFs with the spread in the emission scenarios and basal melt methods. This was done by computing the standard deviation between the median estimates of the Amundsen calibration (QR) for the three SSP scenarios (28 mm for QR) and the standard deviation between the median estimates of the four basal melt

methods for each SSP scenario (21 mm for SSP1-2.6 to 31 mm for SSP5-8.5). The spread between ESMs and RFs is thus larger than the spread between the three SSPs and four basal melt methods.

As a final assessment, the RMSE over the Amundsen region was used to rank the historical performance of individual combinations of ESM-RF pairs. The top 10% best-performing ESM-RF pairs have slightly lower estimates for the Antarctic contribution but similar estimates for the Amundsen contribution (Fig. A2). As a result the relative contribution of the Amundsen region increases compared to the total Antarctic dynamics contribution to sea level, as was also visible in the hindcasts of the top 10% models (Table 7).

To summarise, this assessment of individual models shows that modelling uncertainties of ESMs as well as ice sheet models are a greater source of uncertainties in Antarctic mass loss projections than the emission scenarios and the basal melt computation methods applied in this study. The uncertainties associated with the ocean temperature evolution from ESMs is even larger than those from ice sheet models, despite the bias adjustment that has been applied to the subsurface temperatures. We also find some relations between historical model performance and future projections, which point at model selection as a potential
next step to better understand the future contribution of Antarctic dynamics to sea level changes.

## 4    Discussion

In this study, projections of the sea level contribution of the AIS and the Amundsen region are presented that were calibrated on four decades of ice discharge observations. Calibration was applied on the basal melt parameterisation. The contribution of Antarctica's ice discharge to sea level changes is computed using ocean forcing from state-of-the-art ESMs from Coupled
Model Intercomparison Project Phase 6 (CMIP6) applied to linear response functions from LARMIP-2 ice sheet models. The major strength of this method is that multiple climate and ice sheet models can be combined to assess their full range of modelling uncertainties. A drawback of the method is that non-linearities between thermal forcing and ice sheet mass loss, related to ice sheet instabilities and ocean dynamics, are not considered because we use the linear response functions framework.

Consistent with Levermann et al. (2020), the ocean sectors in our study are somewhat wider than the continental shelf. The advantage of a wider region is that it allows for more assimilated observations in the reanalysis product that is used for the bias adjustment of ocean temperature (the continental shelf region is only sparsely sampled). Furthermore, it should also be noted that we used basal melt anomalies and not absolute basal melt in the computation and calibration of the sea level contribution. This is because that allows us to better represent observed melt but the downside is that anomalies are a second order effect that
is harder to model and observe. We also remark that the linear response functions are derived from ice sheet model experiments with an homogeneous basal melt increase over each entire ice shelf. Therefore, apart from the five regions for which the linear response functions were derived, no spatial patterns and effects are taken into account.

The inability of our models to represent the observed acceleration (Fig. 6) could be explained by ice sheet/ocean feedbacks that are not represented in the models. Recent studies suggest a positive feedback between ice sheet melting and subsurface
ocean warming (Bronselaer et al., 2018; Golledge et al., 2019; Sadai et al., 2020) that could explain this deficiency in the

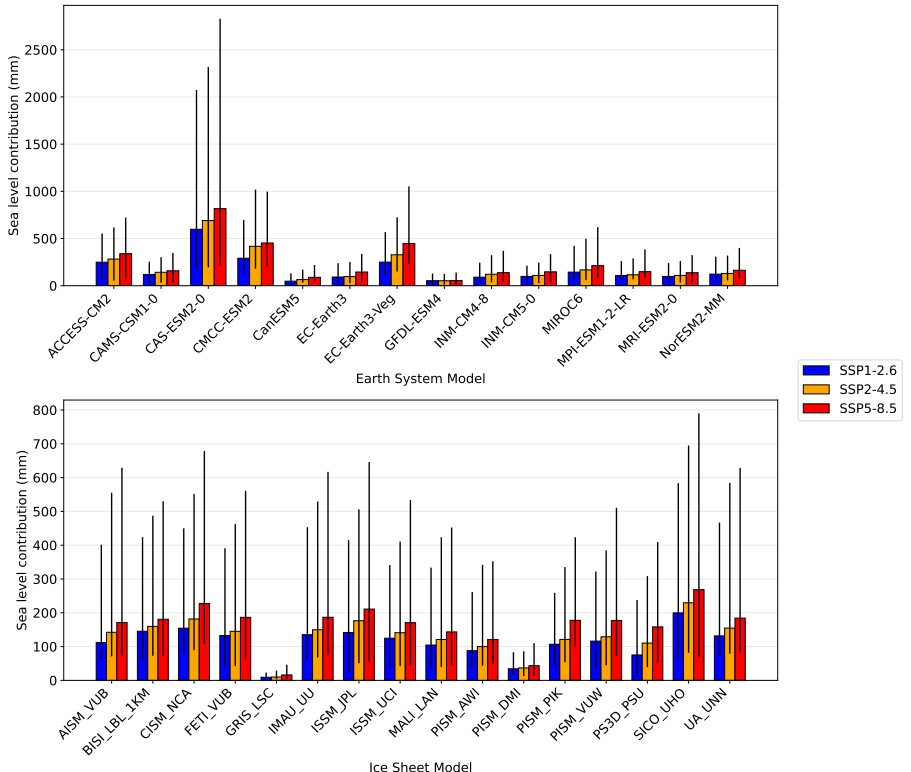

**Figure 12.** Projected Antarctic sea level changes for SSP1-2.6 (blue), SSP2-4.5 (orange) and SSP5-8.5 (red) over the 21st century, defined as the difference between year 2100 and the period 1995-2014. The top panel shows the projections for each CMIP6 ESM, where the errorbars indicate the 17th to 83rd percentiles (computed from the associated RF timeseries). The bottom panel shows the projections for each RF, where the errorbars indicate the 17th to 83rd percentiles (computed from the associated ESMs). Basal melt is computed with the quadratic parameterisation which is calibrated on the Amundsen region (QR). Note the differences in the vertical scale.

models. One reason to introduce the quadratic parameterisation was to account for the observed non-linear relation between ice melt and ocean forcing (Jenkins et al., 2018). However the feedback between surface freshening due to meltwater release, subsurface warming, and enhanced basal ice shelf melt is not represented by this parameterisation. It should also be noted that our study does not address the impact of surface melt on calving nor marine ice cliff instability processes that would lead to higher projections.

In the current generation of ESMs (CMIP6) ice shelf cavities are not represented, leading to deficiencies in the representation of ocean currents and ice-ocean interactions (Mathiot et al., 2017). Including ice shelf cavities in ESMs would better resolve how the inflow/ambient temperature is affected by mixing with meltwater and ocean dynamical processes inside the cavity. Also, the resolution of most CMIP6 ESMs is not high enough to resolve the ocean circulation on the continental shelf, including the Antarctic Slope Current (Thompson et al., 2018). This could lead to a mismatch between observed and simulated ocean warming in the coastal regions. Due to these ocean model deficiencies, temperature-melt relations are typically parameterised

(Favier et al., 2019). We have chosen to use a simple quadratic scaling with far-field thermal forcing (Eq. 3), which could be calibrated on the heat exchange velocity $\gamma$ and applied to all models. This parameterisation performs relatively well when compared with ocean-ice sheet coupled simulations (Favier et al., 2019). The quadratic relation between ice shelf basal melt

and thermal forcing is also confirmed by ocean model experiments (Holland et al., 2008) and observations (Jenkins et al., 2018).

Calibration of the $\gamma$ value in the basal melt parameterisation results for 10-17% of the ESM-RF pairs in a value of zero, which means that, in some cases, the calibration method is invalid. However, we found that each ESM could lead to a successful (positive $\gamma$) calibration if combined with several RFs, so it is the ESM-RF combination which determines whether calibration is

successful. Unsuccessful calibration occurs when the ESM produces large historical natural variability, and the lagged response in the RF translates this into a reduced mass loss over the specific period. In these cases, the ESM produces a weak signal-to-noise ratio in terms of historical warming (the observation period is too short). Overall, the calibration of each ESM-RF pair is dependent on the magnitude and phasing of natural variability (in ocean temperatures and observed mass loss). For the calibrated ESM-RF combinations, the large number of pairs reduces the impact of natural variability on the resultant calibrated

projections.

Calibrating the basal melt parameterisations on observed ice discharge is a way to get more correct historical sea level trends, which was not assessed in ISMIP6. Calibration of individual ESM-RF pairs increased the spread in basal melt sensitivities but decreased spread in the hindcast experiments of Antarctica's sea level contribution. Unfortunately, calibration of the basal melt relation on ice discharge did not reduce the spread in future projections of the ice dynamics contribution to sea level compared

to using observation-based basal melt sensitivities. However, the ice sheet models used to derive the response functions could all be biased in the same direction, resulting in a too high or too low sensitivity to changes in basal melt. For example, if the ice sheet models are not sensitive enough to basal melt perturbations, calibration will result in high-biased melt rates to compensate the low-biased sensitivity. In this case, getting the correct historical ice discharge would not give so much confidence that the response to future warming is correct.

In the LARMIP-2 RFs, the sea-level equivalent ice loss is obtained from the changes in the volume above flotation of the ice sheet. In the calibration we assumed that a change in the volume above flotation equals the grounding line ice discharge. Although changes in the volume above flotation of the ice sheet are strongly related to ice discharge across the grounding line, these two variables are not exactly the same. When the ice sheet is grounded below sea level, only part of the ice that moves across the grounding line will contribute to volume above flotation. In the extreme case, when an ice stream is just about at

the flotation limit and very slightly grounded, it could be that its discharge increases and the grounding line retreats, but the sea-level contribution of this is quasi negligible.

To compute projected sea level change, we have made the assumption that the calibrated gamma values are constant. There are, however, reasons to assume that basal melt sensitivities will change in the future. In the projections (Fig. 4), especially for SSP5-8.5, all coastal regions, especially the Weddell and Ross sectors, experience a warming signal which is not present

in the historical period. As the open ocean outside the cavities warms, it could be expected that this warming will at a certain moment also be transported inside the cavities, and contribute there to basal melt and ice discharge. New calibration will then

lead to larger Antarctic-wide basal melt sensitivities. This means that calibrated basal melt sensitivities that link open ocean subsurface temperatures outside cavities to basal melt underneath ice shelves could be time-evolving.

In this study, an Antarctic-wide and regional Amundsen calibration of the basal melt parameterisation have been applied. The relation between thermal forcing and basal melt is more difficult to derive for the full AIS. The reason is that it includes regions in which ocean warming has not been causally linked to changes in ice dynamics as the warming was too small or absent over the historical period. Moreover, calibrating on the Antarctic-wide response strongly underestimates the historical mass loss in the Amundsen region, which accounts for more than 70% of the observed historical sea-level contribution. Therefore, the Antarctic-wide calibration gives information about a lower bound for the future projections: i.e. what would happen if the total AIS would keep the same basal melt sensitivity to ocean warming in the future. The Amundsen region is considered the best region for calibration since it has been shown that the Amundsen mass loss is dominated by ice discharge due to basal ice shelf melting (Pritchard et al., 2012). Previous studies have shown that ice dynamical changes were causally linked to ocean warming during the observational record (Rignot et al., 2019). It could be expected that when ocean temperatures increase and experience similar warming rates in other regions, the basal melt sensitivity will also increase in those regions.

It should also be noted that the quadratic parameterisation does introduce some regional difference in basal melt sensitivity due to its dependence on the absolute temperature, resulting in a lower sensitivity in colder cavities. When the high basal melt sensitivities derived from the Amundsen calibration are applied to the other regions, the resulting basal melt will thus be smaller due to the colder temperatures. The nonlinear relation between melt and temperature change found in observations (Jenkins et al., 2018) suggests that the quadratic relation based on the Amundsen region might be applicable to the cold-water sectors, although individual regions might still respond differently to similar forcing due to differences in ice and ocean dynamics and ice geometries. The Amundsen calibration is therefore considered more reliable for future projections of the total AIS than the Antarctic-wide calibration, even though it overestimates the total Antarctic contribution to sea level over the historical period.

Remarkably, the projections of emulated ISMIP6 and LARMIP-2 as presented in IPCC AR6 are higher than their counterparts using median basal melt sensitivities of ISMIP6 AntMean and LARMIP-2 (QM and LM, respectively; Fig. 8) in our study (Table 8). The differences between the AIS projections using our methodology with median LARMIP-2 sensitivities and the LARMIP-2 AR6 results can be attributed to differences in thermal forcing since the median basal melt sensitivity and ice sheet response (the RFs) are equal. This means that the thermal forcing in LARMIP-2 AR6 is higher than the forcing in our study, which could be related to the different methodology for the ocean forcing. In the original LARMIP-2 setup, GSAT is used as a driver of the method compared to using bias-adjusted Southern Ocean temperatures in our study. The ISMIP6 emulator uses a joint distribution of the AntMean and PIGL basal melt sensitivities (Edwards et al., 2021). The ensemble mean basal melt sensitivity in emulated ISMIP6 (ca. 10.6 m yr $K^{-2}$) is therefore higher than the median AntMean basal melt sensitivity in ISMIP6 (2.6 m yr $K^{-2}$, Tab. 5), but also higher than our Amundsen calibrated sensitivity (3.7 m yr $K^{-2}$). Based on this higher sensitivity, it could thus be expected that the ISMIP6 emulator gives higher projections than our Amundsen calibration projections. This is not the case, indicating that differences in the methodology other than the basal melt sensitivity lead to a lower ice sheet mass loss in emulated ISMIP6. Part of lower projections can be attributed to the selection of ice sheet models, since the LARMIP-2 median sea level contribution reduces by 0.02–0.03 when using only the 13 ice sheet models common to

ISMIP6 and LARMIP-2 (Fox-Kemper et al., 2021). Another difference with LARMIP-2 and emulated ISMIP6 is that we used a different set of ESMs, which can lead to large differences in the modelled response (see Sect. 3.3.3). These large intermodel differences in ESMs point at model selection as a promising next step to reduce uncertainties in future projections of the con-
tribution of ice dynamics to sea level changes. Since we only used temperature anomalies from ESMs as forcing, the selection criteria should not be based on the mean climate but on climate trends. The methodological differences with ISMIP6 AR6 are even larger than for LARMIP-2 since ISMIP6 does not use the linear response functions framework but runs offline ice sheet models to account for the ice sheet response. Despite all these differences in methodology, we arrive at projections which are in line with previous multi-model assessments of the dynamic contribution of Antarctic mass loss to future sea level.

## 5  Conclusions

This study presents calibrated projections of the contribution of Antarctica's ice discharge to sea level in 2100 compared to present-day (1995-2014). Since there is still high uncertainty in the temperature-basal melt relation (Dinniman et al., 2016), we applied a new approach to constrain this relation (Fig. 1). This was done by calibrating the modelled response on ice discharge observations rather than observation-based estimates of basal melt. The new projections of the sea level contribution are therefore constrained by historical ice discharge observations of the Amundsen region and the total Antarctic ice sheet (Rignot et al., 2019). Ocean thermal forcing is based on regional subsurface ocean temperature from 14 CMIP6 ESMs and 3 SSP scenarios and bias-adjusted with GREP ocean reanalysis data. The changes in ice discharge are calculated with 16 linear response functions (RF) based on ice sheet model experiments from LARMIP-2 (Levermann et al., 2020).

The results show that a large part of the calibrated basal melt sensitivities are higher than those derived from melt obser-
vations, which is related to a wider spread in the calibrated basal melt sensitivities. The median basal melt sensitivities from calibration on ice discharge are for the Amundsen (Antarctic-wide) calibration higher (lower) than the median values applied in LARMIP-2 and the AntMean method of ISMIP6, but lower than the median value in the PIGL method of ISMIP6. The Amundsen calibration performs better in simulating the sea level acceleration and the dominance of the Amundsen region over the historical period compared to Antarctic-wide calibration, and performs arguably better than the Antarctic-wide calibration
when it comes to future projections (Sect. 4). However, even with calibration on past ice discharge, the acceleration of the sea level contribution during the observational period is underestimated for the Amundsen region, indicating missing physics. Also the relative contribution of the Amundsen region to the AIS sea level contribution is underestimated, but it improved by using a quadratic rather than a linear relation between thermal forcing and basal melting.

For the Amundsen region, the basal melt method impacts the sea level contribution more than the SSP scenarios, whereas for
the total AIS the SSP scenarios become more influential by the end of the 21st century. However, differences related to the SSP scenarios and our methodological choices in the calibration and basal melt computation are small compared to the uncertainties associated with ESMs and RFs. Uncertainties associated with the ocean temperature evolution from ESMs is even larger than those from the RFs of ice sheet models, despite the bias adjustment that has been applied to ocean temperatures. Furthermore, we find that the depth of thermal forcing has a large influence on the resulting sea level contribution in future projections. In

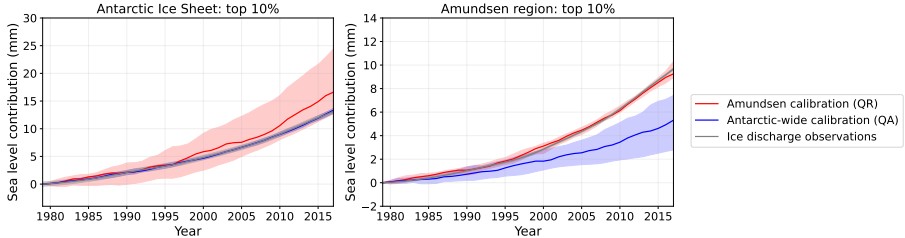

**Figure A1.** Similar as Fig. 6, but for top 10% best-performing ESM-RF pairs.

our study we applied the same thermal forcing depth as in Levermann et al. (2020), which is the forcing near the ice shelf base. Using a thermal forcing depth near the ice shelf base rather than the deepest ocean layer above the continental shelf leads to a larger relative contribution of the Amundsen region to the total Antarctic sea level contribution, which is closer to observations.

The calibration shows that the two main studies on which the IPCC AR6 Antarctic sea level contributions are based (emulated ISMIP6 and LARMIP-2) use median basal melt sensitivities that are higher than the median Antarctic-wide calibrated value that we found. In line with this result, projections of emulated ISMIP6 and LARMIP-2 are higher than projections using our Antarctic-wide calibration on ice discharge. The Amundsen calibration results in median estimates for the dynamic sea level contribution of 0.12 m for SSP1-2.6, 0.14 m for SSP2-4.5 and 0.17 m for SSP5-8.5, sitting in between the emulated ISMIP6 and LARMIP-2 projections, as presented in IPCC AR6 (Fox-Kemper et al., 2021, their Tab. 9.3). Compared to the median basal melt sensitivity from the Amundsen calibration, the basal melt sensitivities in LARMIP-2 are lower but for the ISMIP6 emulator the ensemble mean basal melt sensitivity is higher (Edwards et al., 2021). Interestingly, LARMIP-2 AR6 gives higher projections than our projections with the Amundsen calibration, whereas emulated ISMIP6 gives lower projections than our projections with the Amundsen calibration. This indicates that methodological differences between our study and emulated ISMIP6 and LARMIP-2 other than the basal melt sensitivity dominate the differences in the dynamic ice sheet mass loss.

## 6   Code and data availability

– Linear response functions from LARMIP-2 (Levermann et al., 2020): https://github.com/ALevermann/Larmip2020/tree/master/RFunctions

– Global ocean reanalyses: https://resources.marine.copernicus.eu/product-detail/GLOBAL_REANALYSIS_PHY_001_026/INFORMATION

– Antarctic ice discharge (Rignot et al., 2019): https://www.pnas.org/doi/suppl/10.1073/pnas.1812883116/suppl_file/pnas.1812883116.sd01.xlsx

– Other code available from reasonable request to the author.

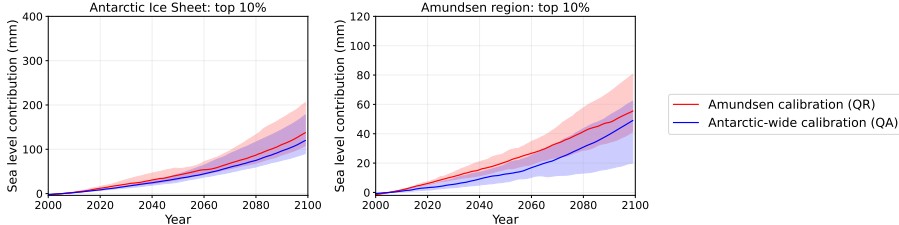

**Figure A2.** Similar as Fig. 7, but for top 10% best-performing ESM-RF pairs.

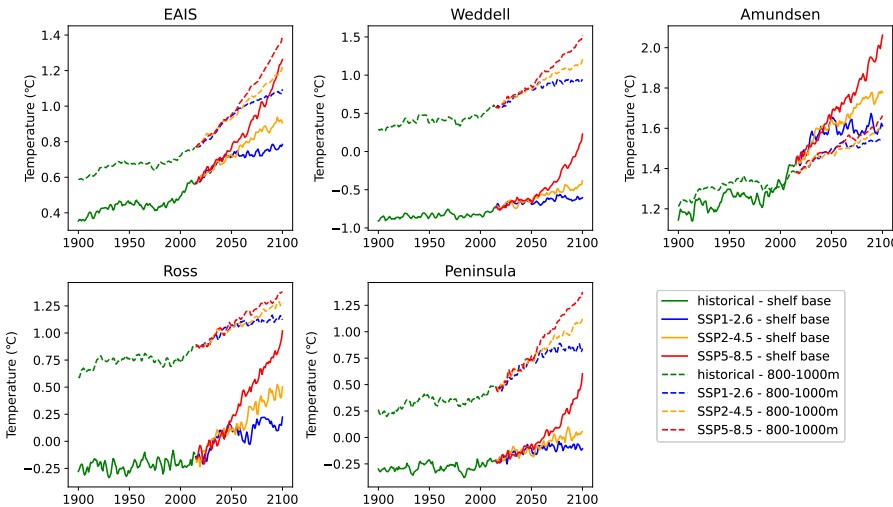

**Figure A3.** Annual mean subsurface ocean temperature time series of the CMIP6 multi-model mean, model drift- and bias-adjusted, for temperatures centered around the mean depth of the ice shelf base (solid lines) and temperatures between 800-1000 m depth (dashed lines).

*Author contributions.* EvdL, SD and DLB designed the study. DLB downloaded the CMIP6 data from the ESGF node and wrote the code to read it. EvdL performed the computations and prepared the manuscript with contributions from all co-authors.

*Competing interests.* The authors declare that they have no conflict of interest.

685 *Acknowledgements.* We acknowledge the editor and two anonymous reviewers for their constructive comments which improved the quality of the paper. This publication was supported by the Knowledge Programme Sea Level Rise which received funding from the Dutch Ministry of Infrastructure and Water Management. This publication was supported by the project RECEIPT (REmote Climate Effects and their Impact on European sustainability, Policy and Trade) which received funding from the European Union's Horizon 2020 Research and Innovation

Programme under Grant agreement no. 820712. This publication was supported by PROTECT. This project has received funding from the European Union's Horizon 2020 research and innovation program under Grant agreement no. 869304, PROTECT contribution number 30.

690

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
