# Peer review of "Antarctic contribution to future sea level from ice shelf basal melt as constrained by ice discharge observations"

_The Cryosphere, 2021_

## Author Comment (AC1)

We would like to thank the editor and the two anonymous reviewers for their effort to review our manuscript and greatly appreciate their helpful comments for improving our study. We will address the raised points during the revision and provide detailed replies to all comments below, with our responses indicated in blue.

***Comment on tc-2021-348***
***Anonymous Referee #1***

*Referee comment on "Calibration of basal melt on past ice discharge lowers projections of Antarctica's sea level contribution" by Eveline C. van der Linden et al., The Cryosphere Discuss., https://doi.org/10.5194/tc-2021-348-RC1, 2021*

*In the manuscript, van den Linden et al. use linear response functions from Levermann et al. (2020) which were derived from ice sheet model responses in sea-level contribution to perturbations by uniform sub-shelf melt rate increases. Sea-level projections are updated by using CMIP6 instead of CMIP5 models and by recalibrating the sensitivity of melt rates to ocean temperature changes using observed mass changes. The authors conclude that with the new calibration, sea-level projections are lowered in comparison to LARMIP2 (Levermann et al., 2020) and ISMIP6 (Seroussi et al. 2020; Edwards et al. 2021).*

*First of all, I want to thank the authors for this well written manuscript which is easy to understand and follow. Unfortunately, I think that the approach presented in the manuscript cannot be applied this way. However, the approach could be interesting and informing further studies, so I suggest two possible modifications that would make it applicable in a methodologically correct way.*

***Major comment:***
*The central issue is that the calibration factor gamma, which relates ocean temperature changes at depth to sub-shelf melt rates changes, is fitted over the historic period in the Weddell Sea, Ross Sea and in East Antarctica, all three regions where very little mass gains or losses have been observed (see Rignot et al. 2019; Figure A1 in the manuscript), in particular in comparison to the overall volume of the regions (if the methodology can be applied to the Antarctic Peninsula should also be checked). This makes a sound calibration with changes in ocean temperatures from CMIP6 models over the historic period basically impossible, due to a number of issues: (1) the changes in mass in the respective region might not even be causally linked to ocean forcing but explainable through, e.g., surface mass balance changes; (2) changes in ocean temperatures in CMIP6 models show a wide spread and how close they are to real changes and if they can actually capture subtleties in the historic record that can be linked to the small changes in ice discharge, is questionable.*

*This means that the calibration factor is fitted between two numbers that are zero or quite small, but have large uncertainties, so that in the end the calibration factor is not really constrained. And this is physically correct, because if there is no enhanced ice discharge due to changes in ocean forcing in the historic record, as for example in the Weddell Sea, the melt sensitivity to ocean forcing cannot be deduced from observations. This problem shows for example in the result that most calibration parameters are zero in many icesheet-*

*ocean-model combinations (section 3.1). And that, even if the parameters are fitted to represent past discharge, they largely underestimate the observed mass loss (Fig. 6).*

*Two suggestions to avoid this problem are:*
*1) focus the study on the Amundsen Sea (potentially also the AP), where actually enhanced ice discharge has been documented extensively and linked to enhanced ocean driven melting. This would allow you to derive a sound fit for that region. Then you could compare the projections for the Amundsen Sea with ISMIP6 and LARMIP2 for that region. Alternatively,*
*2) if you want to include the whole of Antarctica, you could use your proposed method to fit gamma in the Amundsen Sea (and AP), where substantial discharge occurred, and assume an uncertainty distribution of gamma (based on LARMIP, your fit, ISMIP6 calibration,..) for the other regions.*

We would like to thank the reviewer for this constructive criticism. We agree that our results show that the calibration factor gamma cannot be well constrained in regions where historical ice discharge is small. As we discuss in the text and as shown in the titles of the panels in Figure 5, the gamma values for the Ross, Weddell and Peninsula regions are positive for less than half of the model pairs. In contrast, for the Amundsen and EAIS regions, around 80% of the model pairs have a positive calibrated gamma value. In these two regions, the historical period shows a warming in the multi-model mean subsurface ocean temperature (Figure 3) and an increased ice discharge in the observations (Figure A1). Since only for the Amundsen region past ice discharge has been linked to ocean warming, we agree that regional calibration on the Amundsen Sea would be most sound. In the first version of the manuscript, we applied a model selection to circumvent this issue of models that could not be calibrated, but we agree that the proposed solution by the reviewer has a more solid physical basis. Therefore, we will only apply the regional calibration to the Amundsen Sea region. In this way, we are certain that ocean forcing is the main driver and that ice discharge signal is large enough to obtain a sound calibration. We will include projections for the Amundsen region in the body of the manuscript and include a comparison with LARMIP and ISMIP6 for the Amundsen region.
Following the second suggestion of the reviewer, we will also apply these Amundsen calibrated gamma values to the other regions of the Antarctic ice sheet for the quadratic relation. This will lead to lower melting sensitivities than in the Amundsen region since by its definition the quadratic relation implies that sectors that are melted by warmer waters are more sensitive than colder sectors. The nonlinear relation between melt and temperature change found in observations (Figure 5, Jenkins et al. 2018, Nature Geoscience) also suggests that the quadratic calibration based on the Amundsen region might be applicable to the cold-water sectors.

In addition to the Amundsen calibration, we believe that an Antarctic-wide calibration could still be interesting to apply as a lower bound for the future projections. This Antarctic-wide calibration will provide information on what would happen if the total Antarctic ice sheet would keep the same basal melt sensitivity to ocean warming in the future. Calibrating on the total Antarctic ice sheet also means that all regions will be sensitive to future warming and no single region will have a gamma of zero. This could be considered as a low-end scenario since it includes regions with very small ice discharge during the calibration period,

which are expected to melt in the future as the climate warms. We will apply both the linear and quadratic relation for the Antarctic-wide calibration, so that we can compare our results with Levermann et al. (2020).

Another improvement will be that for both the regional Amundsen and Antarctic-wide calibration we will only select the model pairs that have a calibrated gamma value greater than zero. Models that could not be calibrated (i.e. with a gamma of zero) will not be used for the projections.

*Since these would mean major changes to the manuscript and potentially the central findings, in the following I mostly omitted comments on the methodology and results that will be affected by the major comment above:*

***Minor comments:***
*the study uses "discharge" in some places, but what it actually means is "changes in discharge", as the latter should in principle be associated to changes in sub-shelf melt. Please check and correct.*

We agree, this will be corrected.

*page 1, line 24: specify "long-term"*

We mean thousands of years. This will be specified.

*page 2, line 53-55: ISMIP6 used more than the quadratic calibration, and those were two options for calibration (not done at the same time)*

Thanks for pointing this out, we will correct this.

*page 3, line 69, line 85: this is discussed as if the scaling factor in LARMIP only has disadvantages, but it actually has the advantage that also uncertainties in the global mean temperature changes (and not only the CMIP trajectories) were included in the uncertainty estimate*

We will include this advantage of using GSAT in the text.

*page 5, line 100; Table 2: the sub-surface temperatures used as ocean forcing look too shallow for me. The relevant water masses are those at depth of the continental shelf that drive the melting close to the grounding lines. I would expect these more around 800 to 1000m depth.*

We used the same depth values as in Levermann et al. (2020), which are the mean depths of the ice shelve undersides. Ocean circulation characteristics, such as cavity circulation, are not explicitly included in this approximation, although the quadratic parameterisation attempts to account for this. To test how our results depend on the depth, we will repeat the computations with 800-1000m depth ocean temperature data and discuss how this influences our results in the manuscript.

*page 6, line 118-120: why do you not correct for individual model biases but by the ensemble mean?*

We actually do correct the individual models with the ensemble mean of the *reanalysis* products. We will clarify this in the text.

*Fig. 3 caption: change to "Annual… time series of the CMIP6 multi-model mean (green) , model drift and bias-adjusted, and the GREP ensemble mean (orange). Both are smoothed by a five year running average filter."*

This sentence will be rephrased.

*page 7, line 121: this sounds weird, the water is not warmed in the cavities, but it's the change in the pressure that lowers the freezing point and increases the thermal driving*

What we mean to say is that the water in the cavities becomes warmer over time due to inflow of warmer water, but we understand the confusion. We will rephrase the sentence.

*page 6, line 137: that T_f is a constant for each region is a very coarse assumption, could be discussed in the discussion.*

We will discuss this assumption in the discussion.

*page 9, line 171-174: why this condition for the "unbounded"? Would you get a better fit for the Amundsen Sea if you would remove this?*

It should be noted that the best models for the Amundsen Sea are not selected, only the best models for the Antarctic summed response (Figure A2). By applying model selection on the Amundsen region we will get a better fit for the Amundsen Sea. To avoid confusion, we will test if this upper bound has a significant impact, but since it only applies to a few models it will probably not have a large influence on the multi-model mean response.

*section 3.2: LARMIP2 did not calibrate the melt-factor with observed changes in ice discharge, but it did compare the obtained mass losses over the historic period, which actually look like a much better fit than your results (Fig 6, Levermann et al. 2020).*

It should be noted that our Figure 6 shows the results of all model pairs, including those that could not be calibrated (gamma = 0), resulting in an underestimation over the historical period. We will replace this Figure by one that includes only models that could be calibrated (gamma > 0).

*section 3.1: do you have an estimate on the uncertainty of your calibrated melt factors for each combination?*

We do not have error estimates yet for each ESM-RF combination, but it is a good idea to include these. We will add an error estimate for the calibration based on the error estimate

in ice discharge observations by calibrating discharge on the observed discharge +/- one standard deviation error in ice discharge for each ESM-RF pair.

*figure 5: instead show the sensitivity in m/a/K, your legend is hard to see, maybe increase the intensity of the colors?*

Thanks for pointing this out, we will increase the color intensity.

*page 13, line 239: but they should, by construction. This indicated the underlying problem with the methodology.*

We agree that this shows that calibration is not possible in all regions. Therefore, we will follow the suggestion of the reviewer by applying the regional calibration only on the Amundsen region and include only the models that could be calibrated (about 80% of all model pairs).

*discussion: you could add Payne et al. 2021 for a comparison between CMIP5 and CMIP6 effects on AIS sea-level projections; you should discuss the errors that arise through not including surface mass balance changes in your fitting procedure.*

We will add a short discussion on AIS sea-level projection based on Payne et al. 2021. However, it should be noted that we do not aim to reproduce total mass loss, but purely ice discharge (without surface mass balance changes). This was also our motivation to choose the Rignot dataset for fitting, since it includes pure ice discharge. Therefore, our fitting procedure does not include an error due to surface mass balance changes.
* * *
*Comment on tc-2021-348*
***Anonymous Referee #2***
*Review of van der Linden et al. "Calibration of basal melt on past ice discharge lowers projections of Antarctica's sea level contribution".*

*General Comments:*

*In this paper, the authors use linear response functions of ice sheet models from LARMIP-2 to estimate contributions to sea level in three future emissions scenarios. Forcing is derived from ocean conditions in CMIP6 models and is further bias-adjusted and corrected for model drift. Their methodology resembles that of LARMIP-2, however the key difference is that instead of calibrating basal melt rates on observed melt rates (as in ISMIP6), basal melt rates were calibrated on observation-based estimates of ice discharge (Rignot et al., 2019). The authors use two different forms of melt rate parameterization (linear and quadratic) and then use their derived values for gamma to update sea level rise projections under three SSP projections out to 2100. They conclude that their method results in lower projections of Antarctic sea level contribution that the other main methods used in IPCC AR6 (ISMIP6 and LARMIP-2).*

*I want to thank the authors for this work, and their new approach in calibrating to a different observational dataset than has been typically used in the community so far. They have recognized and tried to address a need to better-constrain gamma values. That said, I have a few broad-brush concerns about the paper.*

*First, the calibration of basal melt rate is based on past ice discharge in regions (e.g. Ross, Weddell) where very little ice loss was taking place. As a result, gamma values are often equal to zero, and so future contributions to sea level from these regions is inevitably muted despite the potential for actual increases in thermal forcing. My concern here is that it is hard to derive a meaningful value for gamma in regions where there is minimal mass loss, and therefore hard to deduce the actual sensitivity of the region to changes in ocean temperature. The uncertainty around CMIP6 ocean warming coupled with the lack of historical mass loss in some of these key regions makes finding a valid range of gamma very difficult. This problem is manifested when the authors find that ESM-RF pairs underestimate the magnitude of observed Antarctic sea level response despite being tuned to match it. Furthermore, after picking the top 10% of ESM-RF pairs, some regional hindcasts do not capture the observed sea level response (Figs A1 & A2). As a result, I'm concerned that the conclusions drawn about the future contributions to sea level, and their uncertainty analysis, are rendered less credible.*

We agree that it is difficult to derive gamma values from regions with minimal mass loss over the calibration period. This could indeed lead to an underestimation of the contributions of these regions in future projections. We will therefore adjust our method, by applying regional calibration only on the Amundsen region (in addition to the Antarctic-wide calibration), as proposed by reviewer #1, since this region shows significant mass loss that is mainly caused by ocean forcing. For a more elaborate answer we refer to our reply to the first comment of reviewer #1.

*Second (and this is a correctable issue) I felt that the rational/motivation for doing the calibration on ice discharge rather than basal melt rate was not well-articulated. I think the authors could add stronger language for why this method is worthwhile. This should also be put in the context of ISMIP6 and LARMIP-2 methodologies, and what potential issues calibrating to basal melt rate could create.*

We will specify this better in the text. We will include these two main points: 1) We want to capture the ice discharge acceleration during the historical period which cannot be derived from basal melt itself due to the delayed response of ice discharge to basal melt. 2) ice discharge measurements capture the entire ice sheet through satellite measurements of ice height and velocity and therefore are better constrained than basal melt estimates which are not measured for the full ice sheet and for the full time period that we use for calibration.

*Specific Comments:*

*L41: UK-ESM has included an evolving ice sheet in a GCM framework. And there are other efforts currently in the works in other groups to include an evolving Antarctic ice sheet in a ESM. So not sure what is meant by 'short-term' here I guess.*

We mean that it is not yet the standard CMIP6 setup so ESM setups with coupled ice sheets cannot yet be used in the 'short-term' in multi-model studies. With 'short-term' we mean in the coming years. We will clarify this and shorten the discussion on coupled ice sheet-ESM modelling in this paragraph, since it is not the main topic of the current paper.

*L69: Please specify why GSAT is a less desirable metric than subsurface temp. Or alternatively, why is it a more useful metric? Easier to derive?*

GSAT is easier to derive, but it does not account for (future changes in) Southern Ocean dynamics. Deriving forcing underneath ice shelves from global mean surface temperatures is quite a rough assumption. Southern Ocean dynamics is probably difficult to capture in just a scaling factor and time lag. It could be expected that a regional metric has a better relation with forcing underneath ice shelves (see also Lambert et al., 2021). We will expand the discussion on this issue in the text.

*L72:  How is future loss consistent with past loss? Please elaborate on pros/cons of ISMIP6 calibration methodology and your methodology here. What makes your useful? Motivation needed.*

We mean that the physics should be consistent, under the assumption that no new processes are taking place. So, if we apply a certain melt relation for future projections, it should also be applicable during the historical period. Otherwise, the melt relation is not consistent. Therefore, we apply the same melt relation to the past and the future. If this does not work, it shows that either there is a deficiency in the melt relation or new processes are taking place. We will specify this better in the text.

*Table 2: Is this the mean grounding line depth? I believe so, but if it is, shouldn't Amundsen GL depth be closer to ~500m?*

It the mean depth of the ice shelf base, similar as in Levermann et al. (2020).

*L108 – L114:  Could rearrange this to start with the motivation first (ocean T bias will affect magnitude of basal melt rate in quadratic estimate, therefore these are the steps we take to deal with it).  Currently this feels like you don't know why you're dealing with bias correction until after it is explained.*

Thanks for pointing this out, we will rephrase this paragraph.

*L118:  Why remove the mean bias, which may even have the opposite sign to the model-specific bias? Why not just remove the bias for each model?*

This sentence is not well-written. We mean that we remove the bias for each model using the mean of the *reanalysis* products. We will clearly state this in the text.

*L143: Please note why/how this assumption is flawed.*

The assumption that the inflow/ambient temperature remains constant in the cavity is flawed since it is affected by mixing with meltwater. Also, ocean dynamical processes are not included in this assumption (e.g. the cavity circulation). We will add this in the text.

*L148-149: Is there wide variability in salinity between far-field and sub-shelf? Ie. Please comment on whether using far-field salinity climatology is a fairly broad assumption, or not? Please discuss. The same goes for Tf too.*

Salinity is used to compute the freezing point temperature Tf. We will analyse the sensitivity of Tf to salinity changes and comment on it in the discussion.

*L166: consider rephrasing this since but here you say sea level contributions are calibrated on ice discharge, but the rest of the paper states that basal melt rate is calibrated on ice discharge. I understand what you mean here, but this may be confusing for readers.*

We understand this confusion and will change the sentence to be consistent with the rest of the paper.

*Section 2.3 in general requires more work to increase clarity for the reader. Please describe the iterative process, how it works. And please expand on the rationale for why to do unbounded vs. bounded methods, as well as the discussion on how the 'unbounded' calibration range is determined. The final paragraph of this section also needs some more clarification.*

We discuss these items indeed very shortly and will further elaborate on the methodology.

*L138: Please define T0 . It is not defined.*

It is defined near Equation 1 in Section 2 (Methodology).

*L227: What calibrated gamma values? All of them? The full range? The median? Mean? This is unclear to me.*

The values are shown in Figure 5. We use for each ESM-RF pair the best-fitting value (with the lowest RMSE compared to Rignot ice discharge). This will be specified in the caption.

*L239-242: What is "reasonable extent"? Also, this seems fairly problematic to me. If the ESM-RF pairs cannot capture the magnitude of the observed Antarctic sea level response, even though they are calibrated to do so, what does this say about the methodology? This is assuming that the past conditions and sensitivities hold into the future as well, which may or may not be true, particularly when feedbacks become triggered (e.g. MICI). This is illustrated by the observation that pre-2010 discharge is overestimated, and post-2010 discharge is underestimated.*

Reasonable extent is a subjective term. We believe that if half of the models has a hindcast with a magnitude that is less than half of the observed discharge we should not use them for future projections. Therefore, we apply a model selection in which we only select models

that had the best results in reproducing the past. It could be the case that some models do not resolve the relevant processes such as ocean dynamics. We do indeed assume that past conditions hold in the future since we apply a constant gamma value, although it should be noted that the quadratic relation also depends on the absolute ocean temperature and therefore the basal melt sensitivity will change over time. Our results could lead to the conclusion that this assumption of a constant gamma is not valid, which would also be useful information. However, deficiencies could also be explained by a lack of some (past) feedbacks in the models, such as the freshwater-temperature feedback. This could for example (partly) explain why the discharge acceleration is not resolved well. This has already been discussed in the discussion section, but we will add some arguments in the results section as well.

*Figure 6: What gamma is used? Median? Mean? Full range? Is the shaded area intermodal spread, are you using a spread in gamma too or just a singular gamma value?*

The best-fitting gamma value (lowest RMSE) for each ESM-RF pair. The shaded area is intermodel spread based on the single best-fitting gamma-values for each ESM-RF combination.

*L407: Please spell out the argument that applies to ISMIP6 gamma-values.*

We will combine this sentence with the previous one, since similar arguments apply.

*In general I enjoyed the discussion, and thought that plenty of this type of language could have been used as motivation/context in the introduction. Just something to consider.*

Thank you. We will consider using parts of the discussion in the motivation/context in the introduction and result section.

*Minor Comments:*

*L 7-8: Explicitly state all three SSP scenarios used.*

We will write out the full descriptions.

*L23-24: include references for model (ice sheet models, I assume?) and geological data being referred to.*

We will add a table with the model names (both ice sheet models and CMIP6 earth system models)

*L25: Moreover, melt of Antarctic land ice*

We will add Antarctic.

*L32: Using similar methodologies to what? To each other, I assume…?*

Yes. We will add this in the text.

*L32-35: Can you mention why uncertainties are increasing?*

Yes, we will add this if known.

*L36: Future projections are always based on modeling (not often).*

We will change often to always.

*L38 & L42: What do you mean "balanced by data from ESMs"? Please clarify*

We mean that because they are coupled their boundary conditions cannot be prescribed but are specified through their interaction with the ocean and atmosphere of ESMs (L38). And that observations act as boundary conditions (L42). We will clarify this in the text.

*L39: Is there a citation for claiming ice sheet-ESMs show undesirable/unrealistic trends ?*

We suggest to remove these sentences ('because ice sheet models …. in the short term.") as discussing this topic in detail might not be directly relevant for our paper.

*L46: Used as onebasis for projections…*

We will change 'a' to 'one'.

*L56: In that study, the temperature…*

We will add the comma.

*L75-76:  What does it mean to arrive at an estimate of future mass loss that is consistent with observed mass loss over the past four decades?*

That the physics is consistent (in this case the computation of basal melt).

*L86: Reference Lambert et al (2021) here would be appropriate.*

Indeed, we will add this reference to the sentence.

*L91: Please state explicitly what Levermann et al calibrates to.*

Levermann et al. does not calibrate the basal melt parameterization, but uses melt sensitivities derived from observations of Jenkins (1991) and Payne et al (2007), resulting in a range of 7-16 m/yr/K. The sensitivity is drawn randomly and uniformly from this interval.

*L97: What does "delayed ice sheet response" mean here?*

The delayed response to basal melt.

*Table 1:  What level does "subsurface ocean temperature" refer to here? I assume it is the mean depth of the grounding line (table 2), but please be clear.*

It is the mean depth of the ice shelve underside since the LARMIP-2 linear response functions are forced with a spatially constant basal melt anomaly. We will specify this in the text.

*L103: The ocean temperature time series…?*

Yes, we will specify this.

*L107: Remove text: "For the quadratic melt parameterization".*

OK. We mentioned this because it is only relevant for the quadratic relation, but it does not matter for the linear one so we will remove it from the sentence.

*L123 & 142: Do any CMIP6 models represent cavities?*

Not that we know of.

*L126: Table 3 is referenced, but perhaps prematurely. I suggest T3 and T4 are swapped in order.*

Yes, that makes sense. We will see if we can add the reference to Table 3 at a later moment.

*L138: "See Table 4…" already noted earlier.*

OK, we will remove this sentence.

*Figure 5: Increase font size here please.*

OK.

*L227: "as specified on top" à "as specified in the titles"*

 OK, we will change this in the text.

*L281-285:  This text seems to be repeated again later on from L290-295.  Please give this another proofread to make sure there is no longer repetition and the flow works well.*

We will correct this repetition.

*L326: "The differences with ISMIP6 and LARMIP-2…"  I think you mean difference between this study's results and these two studies, but this phrasing leaves this ambiguous.*

Indeed, we will rephrase this sentence.

*Fig11&12: Label the x-axis please. And possibly increase fontsize.*

We will add a label to the x-axis and increase the fontsize.

*L345: "is computed using forcing from state-of-the-art…"*

We will change 'with' to 'using forcing from'.

*L346: "CMIP6 applied to linear response functions…"*

We will change this.

*L364: Is there an appropriate reference for the comments made about ESM performance?*

 We will add a reference here.

*L373: By climate-state dependent do you mean time-evolving?*

 Yes, depending on e.g. the subsurface ocean temperature.

*Table 7: Please state what the percentages refer to. Also add units.*

Yes, we will add an explanation to the percentile numbers. Units are specified in the caption.

*L376: Please elaborate on the processes you refer to here.*

We will elaborate on the processes.

*L384: The better prediction of future mass loss you do achieve is for less physically defensible reasons, no?*

If the gamma value would be a constant, like we assumed for our study, it would indeed be less physically defensible. However, if it turns out that gamma is time-evolving it could be defensible. For example, if the cold ocean sectors become warmer and their temperature closer to the Amundsen region it could be expected that Antarctic-wide or calibration on the Amundsen region works better than regional calibration on those specific sectors. We will elaborate our discussion on this point.

*L386: Elaborate on the advantages and disadvantages.*

We will elaborate on this.

*L395: Therefore, it is interesting…*

We will add a comma.

*L434-435: Please state what you mean by 'highest' and 'lowest' methods.*

The methods that give the highest and lowest projections. We will specify this in the text.

*FigA1 & A2: Font sizes of tick labels and axis labels could be increased for better readability.*

We will increase the font sizes.

*In general, section headings could be more clear. For example, "Magnitude and Rate" and "Best Estimate" could be more specific to help the reader understand what the section is about.*

We will make the headings more specific.

---

## Referee Report (RR1)

Review of van der Linden et al. " Antarctic dynamics contribution to future sea level constrained by ice discharge observations".

General Comments:

In this paper, the authors use linear response functions of 16 ice sheet models from LARMIP-2 to estimate contributions to sea level under three different SSP future emissions scenarios. Forcing is derived from ocean conditions in 14 CMIP6 models. Different methods tested include trying different depths for ocean thermal forcing, two types of parameterizations (linear, quadratic), and differing regions for basal melt sensitivity calibrations. Instead of calibrated to observed melt rates as in other studies to date, van der Linden and others calibrate to four decades of observed ice discharge (Rignot et al., 2019). This is done for an Amundsen-specific calibration as well as an Antarctic-wide calibration.

I thank the authors for their work to implement comments from the previous round of reviews. I think it is clear there has been a lot of effort in these revisions, and I think the new methodology takes care of the major notes raised in the first round. The authors have now taken care of the issue of what happens when calibrating to a region that has seen very little mass loss in the historical period.

In general, I think the paper reads well now. My main comment is that, as the title states, the paper aims to deliver Antarctic sea level estimates based on a new methodology. However, the way it reads now, the actual SLR values are not woven through the text (the results exist in figures and tables only). I recommend that the Abstract, Results, and Conclusion sections all explicitly state the actual resulting values. This will improve readability as well since, at the moment, it is easy to get lost in all the methods and which ones lead to higher or lower SLR estimates. An example of this might be in lines 365-368, where the authors discuss the contributions they arrive at as compared to other studies (LARMIP and ISMIP6).

Another related issue is the Conclusion section. Plenty of this section is repeated information from the Discussion. I think it would be helpful to spend more time in the Conclusion putting the authors' results in context of other SLR estimates, and again, explicitly stating their new SLR estimates.

Please see below for some other comments that will improve clarity for the reader.

Comments:

Section 2 (Methodology). I think you could use a very brief explanation of the steps in Figure 1 to walk through the steps so the reader knows what to expect in terms of flow before you launch into the details of each step in each subsection that follows.

L105: Why do you choose the depth level 800-1000m in addition to the ice shelf base depth? Is there some rationale to choosing that depth? I know you're trying to see if the depth matters,

but is there a good reason to choose this one? Later on, near line 402, you do mention that this is because this is where relevant water masses that drive melting closer to the grounding line originate, but I'm not sure I exactly follow. At the least a reference is needed here.

Section 2.3
It seems to me that the calibration methods should come before the Sea level contribution method. Because 2.2 ends with talking about calibration, then you start into a very short sea level equation, and the back to the calibration. Consider re-organizing some of this for better flow that matches your actual methodological order.

Minor Comments:

Title. Suggesting: "Antarctic contribution to future sea level as constrained by ice discharge observations"

L36-37: commas: To address this issue, our study aims to gain more insight in the Antarctic contribution to, and uncertainties in, future sea level …

L66: …over ocean temperature changes as a driver is that uncertainties in GSAT…

L69: …this step by using subsurface ocean temperature as the driver ….

L78-79: thereby constraining the basal melt even before the observational period. I think I know what you mean here, but it could be rephrased for clarity.

L79: As a calibration target, ….

L80 & 82: State the year for the Rignot paper explicitly when citing in-line.

L126: Again, I wouldn't say ESM's typically do not represent ice cavities, if none actually do.

L130 and throughout: italicize *in situ*

L148: CMIP6 ESMs do not resolve cavities, as far as I know. Clarify that by removing 'typically'

L188: Do you mean to say the basal melt is computed from subsurface TF anomaly? Also, please state explicitly in this sentence that this is coming from CMIP models.

L208: For the linear parameterizations, we compared our calibrated basal melt sensitivities to the values used in LARMIP-2.

L212-217: Please explain more explicitly why the underestimation/overestimation occurs. I'm not sure the reason is immediately clear.

L218-219: A similar comparison was made for the quadratic paramterization, with the basal melt sensitivities applied in ISMIP6 (Jourdain et al, 2020). Here, the median Antarctic-wide calibrated…

L230: Dataset is one word

L240: "Furthermore, the spread in our calibrated melt sensitivities…"

L243-246: "Models with calibrated melt sensitivity values outside the observation-based ranges would either underestimate or overestimate the past ice discharge if observation-based sensitivities had been applied. As a result, the spread in simulated ice discharge over the historical period will be lower for calibrated basal melt sensitivities than for the observation-based basal melt sensitivities."   I'm not sure the second sentence obviously follows the second. I actually think these sentences are just generally hard to follow, and could use some re-writing to improve clarity.

L280: … with respect to the total Antarctic contribution cannot be reproduced either (about 70%...)

L287: performs better in the other region -- which region??

L292: Nevertheless, the mean response….

L315:  Perhaps indicate that these results are shown in Figure 8 somewhere here.

L322:  There is a discussion of ratios that are higher and lower depending on the method, but please state what the values of these ratios are.

L325: What do you mean by highest basal melt method? Please clarify.

L331: Discussion of large spread in the projections, but please put a value to this.

Table 8: I like this table, and the comparison you draw with ISMIP6 and LARMIP2.  I am wondering why you picked 17% and 83% instead of 5 and 95% (I assume these are percentiles, but you may want to make this explicit in the caption). I also wonder if there is a way of visualizing these results in a figure? At the moment there are plenty of figures showing comparisons of different methods, but it would be nice if there were a figure showing all the final SLR projections as compared to other leading estimates in the literature.

L403:  … grounding line originate from ….

Figure 7:  Could increase size of this figure.

Figure 8 & 9: Please make all 6 panels the same size, and place legend either below or to the right of these six panels. X-axis label on bottom row is missing. Please also indicate more clearly that the blue and orange indicate the main methods used and that pink and yellow indicate the additional test with the single basal melt sensitivity it applied. Also, I think at least in my version, the red line showing the median is difficult to see, particularly in the red and orange distributions. Consider a different color?

L479: …, related to ice sheet instabilities and ocean dynamics, are not considered…

L489: remove (fully)

L502: A physical explanation for a mismatch…. A mismatch in what? Please be specific.

L508: …during the calibration period is representative of the future.

L526: What do you mean it is dominated by ice dynamics? This seems vague.

L545: …uses global mean temperature as the driver…

Consider making Fig 12 & 13 two panels in the same figure since they share the same structure, design, and legend.

Make Figure A1 and A2 the same sizes and ratios. A2 looks more stretched than A1.

L574: remove ' dynamics'

Conclusions section: Much of this is a repeat from the Discussion. Please consider editing the conclusions to include just the biggest take home messages and spend more time putting that in the context of the bigger picture of sea level projections, modeling Antarctic mass loss, and potential avenues and recommendations for future work.

---

## Author Response (AR2)

**We would like to thank the editor and the two anonymous reviewers for their effort to review our manuscript and greatly appreciate their helpful comments for improving our study. We have addressed the raised points and provide replies to all comments below, with our responses indicated in blue.**

**Editor comments**

My minor comments:

- "Ice discharge" is ambiguous, as it may refer to ice discharge at the ice shelf front (calving) or ice discharge across the grounding line (as in Rignot et al., 2019). Please clarify this at the beginning of the manuscript, as well as the link between discharge and sea level.

We added an explanation in the introduction (L. 67-69 & and L. 82-83).

- L. 8-10: there is no verb in this sentence -> are improved ?

Added 'are' (L. 9)

- L. 53-56: it may be worth mentioning Payne et al. (GRL, 2021), which describes the CMIP6 part of ISMIP6.

We added a sentence mentioning Payne et al. (L 61-62).

- "S" is used for both salinity (equ. 4) and sea level (equ. 5, Table 7). Please use different letters or upper/lower case.

Good point. We changed salinity to lower case s.

- L. 165-168: is the uncertainty on Rignot (2019)'s discharge taken into account in the calibration?

It was not included in the calibration since the uncertainty was small compared to the intermodel spread (L. 219-221).

- Tuning the melt rates to get the observed ice discharge is a way to get correct historical sea level trends, which was clearly missing in ISMIP6. However, if the ice sheet models used to derive the response functions are all biased in the same direction, e.g. not sensitive enough to climate perturbations, tuning the melt rates will aim to high-biased melt rates to compensate the low-biased sensitivity. In this case, getting the correct historical ice discharge would not give so much confidence that the response to future warming is correct. I have the same concern with a potential selection of CMIP models based on the observed discharge (as suggested L. 446-448). Please consider discussing this point.
Good point. We added a paragraph on this issue in the discussion section (L. 613-617).

**Anonymous referee #1**

Review on "Antarctic dynamics contribution to future sea level constrained by ice discharge observations"

Main comment:
From the replies I understand now that you use the historic observations of ice discharge to constrain the melt parameter (e.g., p3 lines 80). There is however one discrepancy in units which you might want to consider - I think you should at least discuss it. In the LARMIP2 paper, the ice sheet modelers were asked to provide the results in units of sea-level equivalent ice loss, which is calculated based on changes in the volume above flotation of the ice sheet. This does not directly compare to changes in ice discharge. In the extreme case, when an ice stream is just about at the flotation limit and very slightly grounded, it could be that its discharge increases and the grounding line retreats, but the sea-level contribution of this is quasi negligible.

We did use the grounding line ice discharge from Rignot et al. (2019), which is defined as "ice discharge by glaciers across the grounding line (where ice becomes afloat in ocean waters and detaches from the bed)" which means that is should in principle be comparable to our LARMIP-2 estimates based on the linear response functions which obtain the sea-level equivalent ice loss from the changes in the volume above flotation of the ice sheet. We have added an explanation in the introduction (L. 67-69 & and L. 82-83) and methodology section (L. 82-83).

Minor comment:
- p 1, line 15-16, and p28, lines 587-589: I do not understand your argumentation so far that your results support this statement. If I understand it correctly, it is based on how your calibrated melt parameters compare to the ISMIP6 median parameter. However, the ISMIP6 experiments also included the PIGL calibrations which is much more sensitive, and this is not included in your argumentation, or? So maybe re-calibrating parameters would also reduce the upper range of the ISMIP6 projections based on this parameter calibration, or am I missing something? Please explain your reasoning for this statement better.

Our comparison is based on the ISMIP6 AntMean method, not on the PIGL method. We only used the PIGL method for comparison to our basal melt sensitivity parameters (Fig. 5). If I understand the ISMIP6 set-up correctly, the AntMean method is used for producing their main results as presented in IPCC AR6. We expanded our explanation in the abstract (L. 14-16), results (453-457) and conclusion (L. 703-712).

- p 2, line 35: that the range of uncertainties appears to be increasing is arguably not because the we know less as implied by this formulation, but because more models and processes are included, i.e., the uncertainties become "visible"

Agree. We added this explanation in the introduction (L. 39-40).

- p 3, line 62: I was a bit surprised by calling this a "melt parameterisation" since in my head this is usually a 2-dimensional field of melt rates, but I think this is fine, maybe add a short explanation to make this clear.

We added an explanation in the methodology section (L. 187-189).

- p 7, line 125: this sentence still sounds weird to me as it is not the water that is changing its temperatures in the cavity.

OK, we removed this sentence as it is not strictly necessary for our methodological explanation.

- equation (4) do you also use the ice shelf cavity mean depth when testing the deeper ocean layers?

We use the depth that we use for the thermal forcing: changed in the text (L. 182-183, 186).

- p 9, line 166: show also the equations over which you are optimizing, this would make it easier to understand what you are describing here

We added the RMSE equation (L. 217).

- p 10, 180-181: give the median values, so that they are somewhere in the manuscript.

We added the median basal melt sensitivity values for ISMIP6 AntMean and LARMIP-2 in the caption of Table 3, and L. 385 and L. 387.

- p 10, section 3.1, please show the discharge curves you use for calibration in the Amundsen Sea and for the whole Antarctic ice sheet

The discharge curves are included in Fig. 6 ('ice discharge observations').

- p 10, line 196: if correct, add "..to sea level while CMIP models indicate an increase in ocean forcing,…"

Over the historical period the median of the CMIP6 models does not indicate a clear increase in ocean forcing in all regions (Fig. 4).

- p 10, line 200: add citations that support this attribution

Added citation to Pritchard et al. (2012).

- Fig 5: add % to the numbers in the top of the panels

Done.

- p 13, line 244-246: not sure I understand this sentence, please clarify

We rephrased the text (L295-304).

- p 14, line 255-256: earlier you stated that you include the linear parameterisation for comparison with LARMIP2?

Yes, we make a comparison later with the median basal melt sensitivity from LARMIP2 and therefore use a linear parameterisation. The phrase 'unless specified differently' is a bit vague. We now made it more explicit when the quadratic and linear parameterisation are used (L.312-314).

- p 21, line 384-398: reformulate this argument, you base your reasoning here on a comparison between projections, but we do not know which projection is correct and hence a conclusion about which methodology is better cannot be drawn. Instead you could use papers that support the quadratic relationship (e.g., Holland et al., 2008).

Agree. We included Holland et al. (2008) and Jenkins (2018) to support our argumentation.

- p 24, line 466-467: Please explain more. Which numbers do you compare to conclude this?

We added the standard deviations between the SSP scenarios and basal melt computation methods in the text to quantify the spread and compare it to the spread between ESMs and RFs (L.551-555).

- p 26, line 598: this could also indicate an insensitivity of discharge to basal melt (in the case of no buttressing)

We decided to remove the physical explanation from this sentence as insensitivity is actually not a realistic feature. It simply means that, in some cases, the calibration method is invalid.

- p 26, line 508-514: I am not sure I understand what you mean. Are you basically saying that FRIS cannot be calibrated at the moment?

Yes indeed. We rephrased the sentence so that it becomes clearer (L. 621-622).

- p 26, line 514: it could be misread at the moment that you calibrate with basal melt (not discharge), maybe be clearer here

We rephrased the sentence (L. 628-630).

- p 26: one point that is missing in your discussion is that you consider a constant basal melt rate increase over the entire ice shelf, no spatial patterns and effects are taken into account

We added this point to the discussion (L. 617-619).

- p 26, line 532: "physically correct" – I do not think that you can derive this from your previous reasoning.

Agree. We added citations to support this argument (L. 685).

- p 27, line 559-560: LARMIP2 did not mainly focus on the future, it did compare to historic ice loss and found their projections to be consistent

Although in LARMIP2 a comparison was made with Antarctic mass loss (IMBIE), it was not used as a constraint on the projections. However, since basal melt observations were used as constraint I will remove the comparison with LARMIP2 and ISMIP6 from this sentence.

**Anonymous referee #2**

My main comment is that, as the title states, the paper aims to deliver Antarctic sea level estimates based on a new methodology. However, the way it reads now, the actual SLR values are not woven through the text (the results exist in figures and tables only). I recommend that the Abstract, Results, and Conclusion sections all explicitly state the actual resulting values. This will improve readability as well since, at the moment, it is easy to get lost in all the methods and which ones lead to higher or lower SLR estimates. An example of this might be in lines 365-368, where the authors discuss the contributions they arrive at as compared to other studies (LARMIP and ISMIP6).

We have added the median sea level contributions of our main projections in the abstract, results and conclusion section.

Another related issue is the Conclusion section. Plenty of this section is repeated information from the Discussion. I think it would be helpful to spend more time in the Conclusion putting the authors' results in context of other SLR estimates, and again, explicitly stating their new SLR estimates.

We reduced the Conclusion section and removed repeated information. We also added a paragraph in which we compare our results to LARMIP2 and ISMIP6.

Please see below for some other comments that will improve clarity for the reader.

Comments:

Section 2 (Methodology). I think you could use a very brief explanation of the steps in Figure 1 to walk through the steps so the reader knows what to expect in terms of flow before you launch into the details of each step in each subsection that follows.

We added a brief explanation in the methodology section (L. 109-129).

L105: Why do you choose the depth level 800-1000m in addition to the ice shelf base depth? Is there some rationale to choosing that depth? I know you're trying to see if the depth matters,  but is there a good reason to choose this one? Later on, near line 402, you do

mention that this is because this is where relevant water masses that drive melting closer to the grounding line originate, but I'm not sure I exactly follow. At the least a reference is needed here.

We added an explanation. The deeper ocean layer is chosen as it approximately represents the deeper water masses on the continental shelf that have access to the cavities under the ice shelves (L. 144-145).

Section 2.3
It seems to me that the calibration methods should come before the Sea level contribution method. Because 2.2 ends with talking about calibration, then you start into a very short sea level equation, and the back to the calibration. Consider re-organizing some of this for better flow that matches your actual methodological order.

We first address sea level contribution, since it is used in the calibration. However, we have changed the text now so that 2.2 does not end with calibration and it is only mentioned in the section on calibration.

Minor Comments:
Title. Suggesting: "Antarctic contribution to future sea level as constrained by ice discharge observations"

We have removed 'dynamics' as you suggest, but since we do not focus on the total contribution we have adapted it a bit to: "Antarctic contribution to future sea level from ice shelf basal melt as constrained by ice discharge observations'.

L36-37: commas: To address this issue, our study aims to gain more insight in the Antarctic contribution to, and uncertainties in, future sea level …

Added commas.

L66: …over ocean temperature changes as a driver is that uncertainties in GSAT…

Added.

L69: …this step by using subsurface ocean temperature as the driver ….

Added.

L78-79: thereby constraining the basal melt even before the observational period. I think I know what you mean here, but it could be rephrased for clarity.

Rephrased.

L79: As a calibration target, ….

Added comma.

L80 & 82: State the year for the Rignot paper explicitly when citing in-line.

Added year.

L126: Again, I wouldn't say ESM's typically do not represent ice cavities, if none actually do.

Removed 'typically'.

L130 and throughout: italicize *in situ*

Done

L148: CMIP6 ESMs do not resolve cavities, as far as I know. Clarify that by removing 'typically'

Removed 'typically'.

L188: Do you mean to say the basal melt is computed from subsurface TF anomaly? Also, please state explicitly in this sentence that this is coming from CMIP models.

Added.

L208: For the linear parameterizations, we compared our calibrated basal melt sensitivities to the values used in LARMIP-2.

Added comma.

L212-217: Please explain more explicitly why the underestimation/overestimation occurs. I'm not sure the reason is immediately clear.

We rephrased this paragraph and explained the underestimation and overestimation more explicitly (L. 265-270)

L218-219: A similar comparison was made for the quadratic paramterization, with the basal melt sensitivities applied in ISMIP6 (Jourdain et al, 2020). Here, the median Antarctic-wide calibrated…

Done.

L230: Dataset is one word

Changed data set to dataset.

L240: "Furthermore, the spread in our calibrated melt sensitivities…"

Changed 'the' to 'our'.

L243-246: "Models with calibrated melt sensitivity values outside the observation-based ranges would either underestimate or overestimate the past ice discharge if observation-based sensitivities had been applied. As a result, the spread in simulated ice discharge over the historical period will be lower for calibrated basal melt sensitivities than for the observation-based basal melt sensitivities." I'm not sure the second sentence obviously follows the second. I actually think these sentences are just generally hard to follow, and could use some re-writing to improve clarity.

We rephrased the sentences and made the explanation more explicit.

L280: … with respect to the total Antarctic contribution cannot be reproduced either (about 70%...)

Added either

L287: performs better in the other region -- which region??

The region that was not used for the calibration. Added in text.

L292: Nevertheless, the mean response….

Added comma.

L315: Perhaps indicate that these results are shown in Figure 8 somewhere here.

Added reference to figure in the text.

L322: There is a discussion of ratios that are higher and lower depending on the method, but please state what the values of these ratios are.

We have added the values of these ratios in the text.

L325: What do you mean by highest basal melt method? Please clarify.

We've clarified this in the text.

L331: Discussion of large spread in the projections, but please put a value to this.

We added values for the ratios in the text to quantify the spread.

Table 8: I like this table, and the comparison you draw with ISMIP6 and LARMIP2. I am wondering why you picked 17% and 83% instead of 5 and 95% (I assume these are percentiles, but you may want to make this explicit in the caption). I also wonder if there is a way of visualizing these results in a figure? At the moment there are plenty of figures showing comparisons of different methods, but it would be nice if there were a figure showing all the final SLR projections as compared to other leading estimates in the literature.

We have explained the column names in the captions. We already visualised the projections using the median MIP basal melt sensitivities in Figures 8 and 9. Adding the ISMIP6 and LARMIP-2 AR6 values would make these figures a bit full, therefore we have chosen to put them in the table.

L403: … grounding line originate from ….

Added 'originate'.

Figure 7: Could increase size of this figure.

Done.

Figure 8 & 9: Please make all 6 panels the same size, and place legend either below or to the right of these six panels. X-axis label on bottom row is missing. Please also indicate more clearly that the blue and orange indicate the main methods used and that pink and yellow indicate the additional test with the single basal melt sensitivity it applied. Also, I think at least in my version, the red line showing the median is difficult to see, particularly in the red and orange distributions. Consider a different color?

Done.

L479: …, related to ice sheet instabilities and ocean dynamics, are not considered…

Added commas.

L489: remove (fully)

Removed.

L502: A physical explanation for a mismatch…. A mismatch in what? Please be specific.

Between observed and simulated ocean warming. Added in the text.

L508: …during the calibration period is representative of the future.

Sentence rephrased.

L526: What do you mean it is dominated by ice dynamics? This seems vague.

Replace 'dynamics' with 'discharge due to basal ice shelf melting'.

L545: …uses global mean temperature as the driver…

Added 'the'.

Consider making Fig 12 & 13 two panels in the same figure since they share the same structure, design, and legend.

Done.

Make Figure A1 and A2 the same sizes and ratios. A2 looks more stretched than A1.

Done.

L574: remove ' dynamics'

Done.

Conclusions section: Much of this is a repeat from the Discussion. Please consider editing the conclusions to include just the biggest take home messages and spend more time putting that in the context of the bigger picture of sea level projections, modeling Antarctic mass loss, and potential avenues and recommendations for future work.

Done.

---

## Author Response (AR3)

Dear Nicolas Jourdain,
Thank you for these clarifying and important comments which helped to improve the quality of our paper. We have addressed them all in the revised manuscript. Our replies are indicated in blue.

1- I don't think that you have addressed the main comment of Referee #1. This comment was about the difference between two variables: the ice flux across the grounding line and changes in the volume above floatation. These two variables are strongly related but not exactly the same (as in the example provided by Referee #1, and possibly in case of variations in the surface mass balance). This (minor?) caveat should be mentioned.

Thanks for pointing this out, we misinterpreted this comment in our previous revision. We have added a paragraph in the Discussion (L. 592-598) which mentions this caveat between grounding line ice discharge and changes in the volume above flotation.

2- There are now several mentions to results "presented in the IPCC AR6" with a simple reference to Fox-Kemper et al. (2021). I think this needs to be clarified as chapter 9 of IPCC AR6 contains various numbers related to ISMIP6 (e.g., CMIP5/CMIP6 forced, raw or emulated, including or not the historical dynamical response) and LARMIP2 (all models or subset). You should probably use the exact naming convention that is used in Table 9.3 of this chapter, and explicitly indicate "Fox-Kemper et al. (2021, their Tab. 9.3)". On a similar note, please be more explicit about the SMB contribution that you remove from the ISMIP6 projections (SMB from ISMIP6 or as provided in Table 9.3?).

Good point. We use emulated ISMIP6 (incl. historical dynamic response) and LARMIP-2 (all models). We have added references to Table 9.3 in the text and explain in the caption of our Table 8 exactly which values we use in our comparison. For SMB we have used the values provided in Table 9.3 as explained now in the caption. We also adapted the naming convention of Table 9.3 in our Table 8. Furthermore, we added 'emulated' to ISMIP6 throughout the text, when we make a comparison with the ISMIP6 emulator results.

3- Regarding your response to the minor point of Referee #1 about the AntMean (originally MeanAnt) vs PIGL calibration used in ISMIP6: the upper bound of "ISMIP6 CMIP5-forced" in IPCC-AR6 Table 9.3 does correspond to PIGL (see Seroussi et al. 2020's Fig. 12d) and the emulation of ISMIP6 presented in Table 9.3 does give an important probability for coefficients within the PIGL range (see Fig. 3c of Edwards et al. 2021). I therefore ask you to reconsider this comment and to reformulate your conclusion that you obtain a stronger sensitivity than ISMIP6.

Thank you for this clarification. This does indeed affect our conclusions in the comparison with emulated ISMIP6, since we base our comparison on the basal melt sensitivity parameters. This means that emulated ISMIP6 uses a higher median basal melt sensitivity than the median of the AntMean calibration, and also higher than our sensitivity based on the calibration on the Amundsen region. Despite the higher basal melt sensitivity in emulated ISMIP6, the projections are lower than our projections based on the Amundsen calibration, suggesting that methodological differences other than the basal melt sensitivity explain the differences between our projections and emulated ISMIP6.